# Unifying speed limit, thermodynamic uncertainty relation and Heisenberg principle via bulk-boundary correspondence

Yoshihiko Hasegawa ®[1] ✉

The bulk-boundary correspondence provides a guiding principle for tackling strongly correlated and coupled systems. In the present work, we apply the concept of the bulk-boundary correspondence to thermodynamic bounds described by classical and quantum Markov processes. Using the continuous matrix product state, we convert a Markov process to a quantum field, such that jump events in the Markov process are represented by the creation of particles in the quantum field. Introducing the time evolution of the continuous matrix product state, we apply the geometric bound to its time evolution. We find that the geometric bound reduces to the speed limit relation when we represent the bound in terms of the system quantity, whereas the same bound reduces to the thermodynamic uncertainty relation when expressed based on quantities of the quantum field. Our results show that the speed limits and thermodynamic uncertainty relations are two aspects of the same geometric bound.

The bulk-boundary correspondence is a guiding principle for solving complex and strongly coupled systems[1–3]. The main idea of the bulk-boundary correspondence is that the information on the bulk of a system is encoded in its boundary. In particular, a system that is complex with no apparent approaches for solving problems can be mapped to a different system that becomes simpler to tackle. By using the bulk-boundary correspondence, a strongly correlated quantum field theory (conformal field theory; CFT) is mapped to classical gravity (anti-de Sitter space; AdS) at one dimension higher, where physical quantities in the boundary are evaluated via those in the bulk space[4–6], which is referred to as the AdS/CFT correspondence.

In the present manuscript, we consider quantum and stochastic thermodynamics[7–10]. They are associated with quantities such as heat, work, and entropy that can be defined based on a stochastic trajectory. Stochastic and quantum thermodynamic systems exhibit behaviors that occur far from equilibrium and are described by correlated and coupled Markov processes. This fact leads us to consider that the bulk-boundary correspondence might play a fundamental role in stochastic and quantum thermodynamics. Recently, refs. 11,12 proposed the continuous matrix product state representation that enables the realization

of the bulk-boundary correspondence in Markov processes. The continuous matrix product state relates a Markov process to the quantum field, with the Markov process and the quantum field corresponding to the boundary and the bulk, respectively. Using the continuous matrix product state, we can investigate the properties of a quantum field from the point of view of the corresponding Markov process. In contrast, we can study a Markov process by mapping it to a quantum field and unveiling its properties. Indeed, the continuous matrix product state has been employed in the thermodynamics of trajectory, where it has been used to investigate phase transitions and the role of gauge symmetry in classical and quantum Markov processes[13–15]. Moreover, we have recently employed the continuous matrix product state to derive quantum thermodynamic uncertainty relations[16–18].

In the present paper, we use the bulk-boundary correspondence to examine thermodynamic bounds, such as thermodynamic uncertainty relations[16,19–40] (see ref. 41 for a review) and (quantum and classical) speed limit relations[42–54] (see ref. 55 for a review). The speed limit relation concerns a trade-off relation between the speed of time evolution and thermodynamic costs, and was first introduced in quantum dynamics[42–48]. Recently, the concept has been generalized

---

[1]Department of Information and Communication Engineering, Graduate School of Information Science and Technology, The University of Tokyo, Tokyo 113-8656, Japan. ✉e-mail: hasegawa@biom.t.u-tokyo.ac.jp

to classical Markov processes as well[50–54]. It states that faster time evolution should be accompanied by higher thermodynamic costs, such as dynamical activity and energy. The thermodynamic uncertainty relation gives the fundamental limit for the precision of thermodynamic machines and states that higher precision can only be achieved at the expense of higher thermodynamic costs. Thermodynamic uncertainty relations have become important not only from a theoretical point of view but also from a practical standpoint, such as the estimation of entropy production from measurements[56–60]. As noted above, the continuous matrix product state has been applied to classical and quantum Markov processes. These approaches use the quantum field representation for analyses, but its time evolution has not been explicitly incorporated. In the present manuscript, we introduce a time evolution operator into the continuous matrix product state. The space of the continuous matrix product state is one dimension higher than that of the original Markov process, and the original Markov process exists at the boundary thus, it is referred to as bulk. We apply the concept of the geometric speed limit inequality to the bulk space to derive speed limits [Eqs. (24) and (39)] and thermodynamic uncertainty relations [Eqs. (30) and (40)]. In the resulting speed limit relations, the distances between the initial and the final states are bounded from above by terms comprising classical or quantum dynamical activities. In the case of the thermodynamic uncertainty relations obtained in this work, we show that the precision of an observable that counts the number of jumps is bounded from below by costs composed of classical or quantum dynamical activities. We establish a duality relation in that the speed limit and the thermodynamic uncertainty relation can be understood as two different aspects of the geometric speed limit inequality. Specifically, when we bound the geometric inequality with the quantities in the Markov process, the inequality reduces to classical and quantum speed limits [Eqs. (24) and (39)]. In contrast, the geometric inequality becomes the thermodynamic uncertainty relations [Eqs. (30) and (40)] when we bound the geometric inequality with the quantities in the quantum field. This duality is demonstrated for both classical and quantum Markov processes. We also consider the Heisenberg uncertainty relation in the bulk space to show that the Heisenberg uncertainty relation reduces to the thermodynamic uncertainty relation in the Markov process.

## Results

### Continuous matrix product state

Let us consider a quantum Markov process described by a Lindblad equation. Classical Markov processes are included in quantum Markov processes as particular cases (see Eq. (16)). Let $\rho(s)$ be a density operator of the system at time $s$. We assume that $\rho(s)$ is governed by the time-independent Lindblad equation:

$$\frac{d}{ds}\rho(s) = \mathfrak{L}(\rho(s)) = -i\left[H_{\text{sys}}, \rho(s)\right] + \sum_{m=1}^{M} \mathcal{D}(\rho(s), L_m), \quad (1)$$

where $\mathfrak{L}(\bullet)$ is a Lindblad super-operator, $H_{\text{sys}}$ is the system Hamiltonian, $\mathcal{D}(\rho, L) \equiv L\rho L^{\dagger} - \{L^{\dagger}L, \rho\}/2$ with $L_m$ being the $m$th jump operator (there are $M$ jump operators, $\{L_1, L_2, ..., L_M\}$), $[\bullet, \bullet]$ is the commutator and $\{\bullet, \bullet\}$ is the anti-commutator. Here, we assume that $H_{\text{sys}}$ and $L_m$ are time-independent. Suppose that the dynamics start at $s = 0$ and ends at $s = \tau$ ($\tau > 0$). When we apply a continuous measurement to the Lindblad equation, we obtain a record of jump events, given by

$$\Gamma \equiv [(s_1, m_1), (s_2, m_2), \ldots, (s_K, m_K)], \quad (2)$$

where $K$ is the number of jump events and $s_k$ and $m_k \in \{1, 2, ..., M\}$ specify the time and type of the $k$th jump event, respectively. The record of these jump events $\Gamma$ is termed the *trajectory*. For a given trajectory, $\rho(s)$ is governed by a quantum Markov process referred to

as the stochastic Schrödinger equation. By averaging all possible measurements in the stochastic Schrödinger equation, we can recover the original Lindblad equation [Eq. (1)].

We now consider the bulk-boundary correspondence in the continuous measurement of the Lindblad equation. The bulk-boundary correspondence relates a Markov process to the quantum field, and this correspondence is possible through a representation known as the continuous matrix product state[11,12]. When we apply the continuous measurement to Eq. (1), we obtain a trajectory $\Gamma$ [Eq. (2)]. The quantum field that records the trajectory is defined as

$$|\Gamma\rangle \equiv \phi_{m_K}^{\dagger}(s_K) \cdots \phi_{m_2}^{\dagger}(s_2)\phi_{m_1}^{\dagger}(s_1)|\text{vac}\rangle, \quad (3)$$

where $\phi_m(s)$ is a field operator having the canonical commutation relation $[\phi_m(s), \phi_{m'}^{\dagger}(s')] = \delta_{mm'}\delta(s - s')$; $\phi_m^{\dagger}(s)$ creates a particle of type $m$ at $s$ and $|\text{vac}\rangle$ is a vacuum state. The time evolution of the measurement record and the state of the principal system can be represented by the continuous matrix product state:

$$|\Phi(t)\rangle = \mathfrak{U}(t; H_{\text{sys}}, \{L_m\})|\Phi(0)\rangle, \quad (4)$$

where $\mathfrak{U}(t; H_{\text{sys}}, \{L_m\})$ is an operator parametrized by $t$ and the operators $H_{\text{sys}}$ and $\{L_m\}$:

$$\mathfrak{U}(t; H_{\text{sys}}, \{L_m\}) \equiv \mathbb{T} \exp\left[-i\int_0^t ds\left[H_{\text{sys}} \otimes \mathbb{I}_{\text{fld}} + \sum_m \left(iL_m \otimes \phi_m^{\dagger}(s) - iL_m^{\dagger} \otimes \phi_m(s)\right)\right]\right]. \quad (5)$$

Here the initial state is $|\Phi(0)\rangle = |\psi(0)\rangle \otimes |\text{vac}\rangle$ with $|\psi(0)\rangle$ being the initial state in the system, $\mathbb{T}$ is the time ordering operator and $\mathbb{I}_{\text{fld}}$ is the identity operator in the field. $|\Phi(t)\rangle$ records the jump events within the interval $0 \leq s \leq t$. Figure 1 shows an intuitive illustration of the bulk-boundary correspondence in Markov processes. Figure 1a shows an example of a Markov process, where the horizontal and vertical axes denote the time $s$ and the state of the Markov process, respectively. By using the bulk-boundary correspondence, all information concerning measurement is recorded by creating particles in the quantum field by applying $\phi_m^{\dagger}(s)$ to $|\text{vac}\rangle$. The bulk-boundary correspondence maps the system to a quantum field that is one dimension higher than the original one, as depicted in Fig. 1b. In Fig. 1b, the original time evolution of the Markov process is shown by the $s$ axis while the extra dimension $t$ in the bulk space represents the time evolution of the continuous matrix product state. In Fig. 1b, the boundary at $t = \tau$ represents the original Markov process, and thus the space of Fig. 1b is the bulk space. Any information that can be obtained from the original Markov process can be derived from Eq. (4). Let us define

$$\rho_{\text{sys}}^{\Phi}(t) \equiv \text{Tr}_{\text{fld}}[|\Phi(t)\rangle\langle\Phi(t)|], \quad (6)$$

where $\text{Tr}_{\text{fld}}$ is the trace with respect to the field. $\rho_{\text{sys}}^{\Phi}(t)$ satisfies $\rho_{\text{sys}}^{\Phi}(t) = \rho(t)$, where $\rho(t)$ is the density matrix in Eq. (1). The quantum field that encodes all information about the jump events is given by

$$\rho_{\text{fld}}^{\Phi}(t) \equiv \text{Tr}_{\text{sys}}[|\Phi(t)\rangle\langle\Phi(t)|], \quad (7)$$

where $\text{Tr}_{\text{sys}}$ is the trace with respect to the system. See Supplementary Note 1 for details of the continuous matrix product state.

### Scaled quantum field

We now consider the time evolution of the continuous matrix product state. Since the operator defined in Eq. (5) is already a unitary operator, it seems satisfactory to employ it as its time-evolution

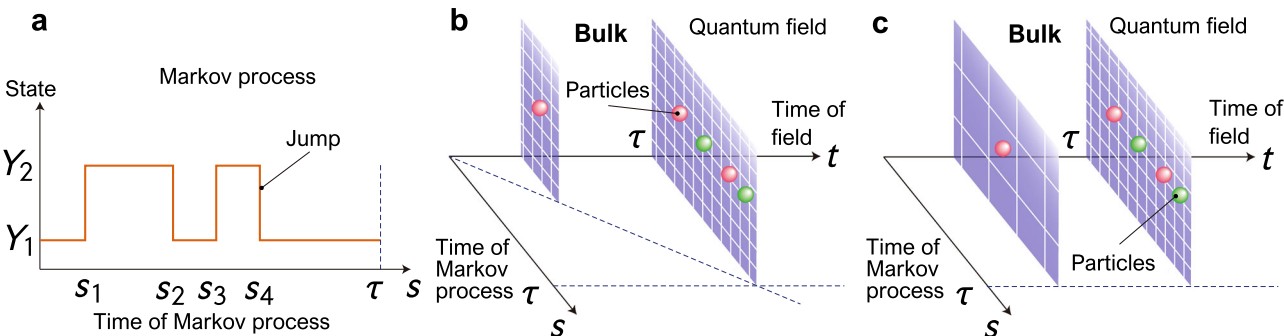

**Fig. 1 | Bulk-boundary correspondence in a Markov process. a** Trajectory of the Markov process as a function of $s$ within the time interval $[0, \tau]$. $Y_1$ and $Y_2$ denote states of the Markov process and $s_k$ is the time stamp of the $k$th jump event. **b** Bulk space corresponding to the Markov process of **a**, generated by Eq. (4). The record of jump events is represented by particle creation in the quantum field. The boundary at $t = \tau$ represents the Markov process of **a**. The axis of $t$ specifies the time evolution of the quantum field. **c** Bulk space corresponding to the Markov process of **a**, generated by Eq. (8). In contrast to **b**, the space is scaled so that the quantum field is defined for $s \in [0, \tau]$ for all $t \in [0, \tau]$.

operator. However, such an approach appears to be problematic, as explained below. We will be interested in the fidelity between two continuous matrix product states at different times, $\langle \Phi(t_2)|\Phi(t_1)\rangle$ for $t_1 \neq t_2$. However, since the integration ranges for $|\Phi(t_1)\rangle$ and $|\Phi(t_2)\rangle$ are different, as indicated by Eqs. (4) and (5), it is not possible to evaluate the fidelity (Fig. 1b). In the present work, instead of using the continuous matrix product state defined by Eq. (4), we employ the scaled representation:

$$|\Psi(t)\rangle = \mathfrak{U}\left(\tau; \frac{t}{\tau}H_{\text{sys}}, \left\{\sqrt{\frac{t}{\tau}}L_m\right\}\right)|\Psi(0)\rangle, \tag{8}$$

where $|\Psi(0)\rangle \equiv |\psi(0)\rangle \otimes |\text{vac}\rangle$. Here, we use $|\Phi(t)\rangle$ and $|\Psi(t)\rangle$ to represent the genuine [Eq. (4)] and the scaled [Eq. (8)] continuous matrix product state representations, respectively. Since $|\Psi(t)\rangle$ and $|\Phi(t)\rangle$ are different states, we show justification for using $|\Psi(t)\rangle$ instead of $|\Phi(t)\rangle$ as follows. Let us define

$$\rho_{\text{sys}}^{\Psi}(t) \equiv \text{Tr}_{\text{fld}}[|\Psi(t)\rangle\langle\Psi(t)|]. \tag{9}$$

In Eq. (8), $H_{\text{sys}}$ and $L_m$ are scaled by $t/\tau$ and $\sqrt{t/\tau}$, respectively, leading to the Lindblad equation $\partial_s \rho(s) = (t/\tau)\mathfrak{L}(\rho(s))$, which is the same as Eq. (1) except for its time scale; the scaled operators yield the dynamics, which is $t/\tau$ times as fast as the original dynamics. Due to the scaling, the integration range in Eq. (8) is the same for all $t \in [0, \tau]$, making evaluation of the fidelity at different times possible. Moreover, the system state (i.e., the state of the original Markov process) can be obtained by both $|\Psi(t)\rangle$ and $|\Phi(t)\rangle$:

$$\rho(t) = \rho_{\text{sys}}^{\Psi}(t) = \rho_{\text{sys}}^{\Phi}(t), \tag{10}$$

where $\rho(t)$ is the density operator in the Lindblad equation (1). Equation (10) shows that, with respect to the state of the system, $|\Phi(t)\rangle$ and $|\Psi(t)\rangle$ provide the state consistent with Eq. (1).

It is helpful to assess the difference between $|\Phi(t)\rangle$ and $|\Psi(t)\rangle$ with respect to a field observable. Let $\rho_{\text{fld}}^{\Psi}(t)$ be a density operator in the field:

$$\rho_{\text{fld}}^{\Psi}(t) \equiv \text{Tr}_{\text{sys}}[|\Psi(t)\rangle\langle\Psi(t)|]. \tag{11}$$

In general, we cannot use $|\Psi(t)\rangle$ instead of $|\Phi(t)\rangle$ for a general measurement in the quantum field. However, if we are interested in the number of jump events, $|\Phi(t)\rangle$ and $|\Psi(t)\rangle$ yield the same statistics since $|\Psi(t)\rangle$ is based on dynamics that are exactly the same as $|\Phi(t)\rangle$ except for the time scale. Since the jump events are recorded in the field as the creation of particles, information of the jump events can be obtained

by measuring the field with the number operator:

$$\mathcal{N}_m \equiv \int_0^{\tau} \phi_m^{\dagger}(s)\phi_m(s)ds, \tag{12}$$

which counts the number of $m$th jumps during $[0, \tau]$. When we are interested in the state of the system (the state of the original Markov process) and the number of jump events, $|\Psi(t)\rangle$ and $|\Phi(t)\rangle$ provide exactly the same information. This property justifies the use of $|\Psi(t)\rangle$ in place of $|\Phi(t)\rangle$.

Thus far, our focus has been on the number operator $\mathcal{N}_m$ alone, but more general observables can be considered. The number operator [Eq. (12)] admits the spectral decomposition:

$$\mathcal{N}_m = \sum_{n_m = 0} n_m \Pi_{n_m}, \tag{13}$$

where the eigenvalue $n_m$ denotes the number of $m$th jumps within $[0, \tau]$ and $\Pi_{n_m}$ is its corresponding projector. The first-level generalization of $\mathcal{N}_m$ is

$$\mathcal{N}_m^{\circ} \equiv \sum_{n_m = 0} \eta_m(n_m)\Pi_{n_m}, \tag{14}$$

where $\eta_m(n)$ is a real function satisfying $\eta_m(0) = 0$. Thus, $\mathcal{N}_m^{\circ}$ is a generalization of $\mathcal{N}_m$ as $\eta_m(n) = n$ recovers $\mathcal{N}_m$ in Eq. (13). The second-level generalization would be

$$\mathcal{N}_m^{\bullet} \equiv \sum_{n_m = 0} \xi_m(n_m)\Pi_{n_m}, \tag{15}$$

where $\xi_m(n)$ is an arbitrary real function. $\mathcal{N}_m^{\bullet}$ is the most general form of observable that commutes with $\mathcal{N}_m$. Note that $\mathcal{N}_m^{\circ}$ and $\mathcal{N}_m^{\bullet}$ can also be used for the scaled representation $|\Psi(t)\rangle$ instead of $|\Phi(t)\rangle$ (see the Methods section).

Figure 1b, c depict the bulk spaces corresponding to $|\Phi(t)\rangle$ and $|\Psi(t)\rangle$, respectively. In Fig. 1c, we see that $|\Psi(t)\rangle$ is defined for $s \in [0, \tau]$, where the scaling factor of the space depends on $t$. In contrast, in the case of Fig. 1b, the quantum field is defined for $s \in [0, t]$ while the scaling factor does not depend on $t$.

## Geometric bound in probability space

The previous section introduced the time evolution of the continuous matrix product state. In this section, we consider the geometric properties of its time evolution. These geometric properties have been extensively employed in the quantum speed limit[55]. We first consider a

space of classical probability and then move to a space of the quantum state in the next section.

Let us consider a classical Markov process with $N_S$ states $\{Y_1, Y_2, \ldots, Y_{N_S}\}$. The dynamics of the Markov process is governed by a classical Markov process:

$$\frac{d}{ds} P(\nu, s) = \sum_\mu W_{\nu\mu} P(\mu, s), \tag{16}$$

where $P(v, s)$ is the probability of being $Y_\nu$ at time $s$ and $W_{\nu\mu}$ is the transition rate from $Y_\mu$ to $Y_\nu$ with $W_{\mu\mu} = -\sum_{\nu \neq \mu} W_{\nu\mu}$. Taking $H_{\text{sys}} = 0$, $L_{\nu\mu} = \sqrt{W_{\nu\mu}} |Y_\nu\rangle\langle Y_\mu|$ and $\rho(t) = \text{diag}([P(\nu, t)]_\nu)$ in Eq. (1), the Lindblad equation is reduced to the corresponding classical Markov process, where $\{|Y_1\rangle, |Y_2\rangle, \cdots, |Y_{N_S}\rangle\}$ constitutes an orthonormal basis with each $|Y_\nu\rangle$ corresponding to $Y_\nu$. Here, the index of the jump operator should be mapped as $L_m \to L_{\nu\mu}$ by mapping $m \to (\nu, \mu)$. Therefore, the $m$th jump in Eq. (1) corresponds to the jump from $Y_\mu$ to $Y_\nu$ in Eq. (16). Using the continuous matrix product state, the probability of measuring a trajectory $\Gamma$ and $Y_\nu$ at the end time is

$$\mathcal{P}(\Gamma, \nu, t) \equiv \langle \Psi(t) | (|Y_\nu\rangle\langle Y_\nu| \otimes |\Gamma\rangle\langle\Gamma|) | \Psi(t)\rangle. \tag{17}$$

Let us consider the time evolution of the continuous matrix product state. Its time evolution corresponds to the $t$ axis in Fig. 1c. Applying the projector $|Y_\nu\rangle\langle Y_\nu| \otimes |\Gamma\rangle\langle\Gamma|$, we can consider the time evolution of the probability distribution $\mathcal{P}(\Gamma, \nu, t)$ as a function of $t$. For such a time-evolving probability distribution, by using ref. 61, the following relation holds:

$$\frac{1}{2} \int_{t_1}^{t_2} dt \sqrt{\mathcal{I}(t)} \geq \mathcal{L}_P\big(\mathcal{P}(\Gamma, \nu, t_1), \mathcal{P}(\Gamma, \nu, t_2)\big), \tag{18}$$

where $\mathcal{I}(t)$ is the classical Fisher information defined by

$$\mathcal{I}(t) \equiv \sum_{\Gamma, \nu} \mathcal{P}(\Gamma, \nu, t) \left( -\frac{\partial^2}{\partial t^2} \ln \mathcal{P}(\Gamma, \nu, t) \right), \tag{19}$$

and $\mathcal{L}_P$ is the Bhattacharyya angle:

$$\mathcal{L}_P\big(p_1(x), p_2(x)\big) \equiv \arccos\big[\text{Bhat}\big(p_1(x), p_2(x)\big)\big]. \tag{20}$$

In Eq. (20), $\text{Bhat}(p_1(x), p_2(x))$ is the Bhattacharyya coefficient:

$$\text{Bhat}\big(p_1(x), p_2(x)\big) \equiv \sum_x \sqrt{p_1(x) p_2(x)}. \tag{21}$$

Here $p_1(x)$ and $p_2(x)$ are arbitrary probability distributions, and Eq. (20) quantifies the distance between the two probability distributions. Equation (18) was used in refs. 51,52 to obtain thermodynamic trade-off relations in classical Markov processes. Note that the probability state in refs. 51,52 is the actual state. This corresponds to $P(\nu, s)$, whose time evolution is the $s$ axis in Fig. 1c. The state considered herein concerns the path probability space $\mathcal{P}(\Gamma, \nu, t)$, whose time evolution is shown by the $t$ axis in Fig. 1c. A straightforward calculation shows that $\mathcal{I}(t)$ can be written as

$$\mathcal{I}(t) = \frac{\mathcal{A}(t)}{t^2}, \tag{22}$$

with $\mathcal{A}(t)$ being the dynamical activity[62]:

$$\mathcal{A}(t) \equiv \int_0^t ds \sum_{\nu, \mu, \nu \neq \mu} P(\mu, s) W_{\nu\mu}. \tag{23}$$

$\mathcal{A}(t)$ quantifies the average number of jumps within $[0, t]$ (see Supplementary Note 2).

The Bhattacharyya coefficient satisfies the monotonicity with respect to any classical channel. Using the monotonicity and Eq. (22), we can write (see Methods)

$$\frac{1}{2} \int_0^\tau \frac{\sqrt{\mathcal{A}(t)}}{t} dt \geq \mathcal{L}_P(P(\nu, 0), P(\nu, \tau)). \tag{24}$$

Equation (24) is the first result of this paper, showing that the distance between the initial and final probability distributions in a classical Markov process has an upper bound comprising the dynamical activity $\mathcal{A}(t)$. Equation (24) is reminiscent of the classical speed limit obtained in ref. 50. The bound in ref. 50 compared the initial and final probability distributions by means of the total variation distance. Equation (24) is a direct classical analog of the geometric quantum speed limit[45].

In Eq. (24), we obtained the lower bound for the right-hand side in terms of the quantity in the system ($P(\nu, s)$ in the Markov process). We next obtain a lower bound using the quantity in the quantum field, which leads to a classical thermodynamic uncertainty relation. We notice that the right-hand side of Eq. (18) can be bounded from below by the distance between $\mathcal{P}(\Gamma, t_1)$ and $\mathcal{P}(\Gamma, t_2)$, where $\mathcal{P}(\Gamma, t) \equiv \sum_\nu \mathcal{P}(\Gamma, \nu, t)$. However, in general, obtaining $\mathcal{P}(\Gamma, t)$ requires a large amount of measurement that is impractical. Thus, as an alternative, we use a time-integrated observable and bound the right-hand side of Eq. (18) with the statistics of the time-integrated observable. Consider the observable in the continuous measurement of the Lindblad equation [Eq. (1)]:

$$\mathfrak{C}(\Gamma) \equiv \sum_m \alpha_m \mathfrak{N}_m(\Gamma), \tag{25}$$

where $\mathfrak{N}_m(\Gamma)$ counts the number of $m$th jumps in a given trajectory $\Gamma$, and $\alpha_m$ is a real parameter defining the weight of the $m$th jump. The Hermitian observable corresponding to Eq. (25) in the quantum field is written by

$$\mathcal{C} \equiv \sum_m \alpha_m \mathcal{N}_m, \tag{26}$$

where $\mathcal{N}_m$ is the number operator defined in Eq. (12). Equation (26) is the weighted sum of jump events during the time interval $[0, \tau]$. Let us define

$$\langle \mathcal{C} \rangle_t \equiv \text{Tr}_{\text{fld}}\big[\rho_{\text{fld}}^\Psi(t) \mathcal{C}\big], \tag{27}$$

$$[\![\mathcal{C}]\!]_t \equiv \sqrt{\langle \mathcal{C}^2 \rangle_t - \langle \mathcal{C} \rangle_t^2}, \tag{28}$$

where $\rho_{\text{fld}}^\Psi(t)$ is defined in Eq. (11). $\langle \mathcal{C} \rangle_t$ and $[\![\mathcal{C}]\!]_t$ correspond to the mean and standard deviation of the number of jump events during the time interval $[0, t]$ in the original Markov process. In Eqs. (14) and (15), we have defined $\mathcal{N}_m^\circ$ and $\mathcal{N}_m^\bullet$, the generalization of the number operator $\mathcal{N}_m$. We also define generalizations of $\mathcal{C}$ as follows:

$$\mathcal{C}^\circ \equiv \sum_m \alpha_m \mathcal{N}_m^\circ, \quad \mathcal{C}^\bullet \equiv \sum_m \alpha_m \mathcal{N}^\bullet. \tag{29}$$

Relations that hold for $\mathcal{C}^\bullet$ should be satisfied by $\mathcal{C}^\circ$, and those that hold for $\mathcal{C}^\circ$ should also be satisfied by $\mathcal{C}$ (see the Methods section).

Applying the inequality relation for the Bhattacharyya coefficient to Eq. (18), we obtain a thermodynamic uncertainty relation for

$0 \le t_1 < t_2 \le \tau$ (see the Methods section for details):

$$\left( \frac{[\![\mathcal{C}^\bullet]\!]_{t_2} + [\![\mathcal{C}^\bullet]\!]_{t_1}}{\langle \mathcal{C}^\bullet \rangle_{t_2} - \langle \mathcal{C}^\bullet \rangle_{t_1}} \right)^2 \ge \frac{1}{\tan\left[ \frac{1}{2} \int_{t_1}^{t_2} \frac{\sqrt{\mathcal{A}(t)}}{t} dt \right]^2}, \qquad (30)$$

which holds for $(1/2)\int_{t_1}^{t_2} \sqrt{\mathcal{A}(t)}/t \, dt \le \pi/2$. Equation (30) is the second result of this paper and holds for an arbitrary time-independent classical Markov process. In refs. 17,37, we derived thermodynamic uncertainty relations that hold for arbitrary classical Markov chains. However, the thermodynamic cost terms in refs. 17,37 are not thermodynamic quantities, whereas the thermodynamic cost in Eq. (30) is the dynamical activity. Let us employ $t_1 = 0$ and $t_2 = \tau$ in Eq. (30). Since there is no jump for $t = 0$, $\langle \mathcal{C}^\circ \rangle_{t=0} = 0$ and $[\![\mathcal{C}^\circ]\!]_{t=0} = 0$, and we obtain

$$\frac{[\![\mathcal{C}^\circ]\!]_\tau^2}{\langle \mathcal{C}^\circ \rangle_\tau^2} \ge \frac{1}{\tan\left[ \frac{1}{2} \int_0^\tau \frac{\sqrt{\mathcal{A}(t)}}{t} dt \right]^2}, \qquad (31)$$

which holds for $(1/2)\int_{t_1}^{t_2} \sqrt{\mathcal{A}(t)}/t \, dt \le \pi/2$. Equations (30) and (31) are previously unknown relations. Note that Eqs. (30) and (31) should hold for $\mathcal{C}$ defined by Eq. (26), since $\mathcal{C}^\circ$ and $\mathcal{C}^\bullet$ are generalizations of $\mathcal{C}$. In addition, Eq. (30) can derive known classical thermodynamic uncertainty relations, as shown below. Let $\varepsilon$ be a sufficiently small parameter. Considering $t_1 = \tau - \varepsilon$ and $t_2 = \tau$ in Eq. (30), Eq. (30) reduces to (see the Methods section for details)

$$\frac{[\![\mathcal{C}^\bullet]\!]_\tau^2}{\tau^2 \left( \partial_\tau \langle \mathcal{C}^\bullet \rangle_\tau \right)^2} \ge \frac{1}{\mathcal{A}(\tau)}. \qquad (32)$$

Equation (32) is equivalent to the bound in ref. 23. Both Eqs. (31) and (32) hold for an arbitrary time-independent Markov process, but the denominator in the left-hand side of Eq. (32) is not the time-integrated observable but rather the time derivative of its average value. The left-hand side of Eq. (31) can be defined through the time-integrated observable $\langle \mathcal{C} \rangle_\tau$, and so can be interpreted as the precision. For the steady state condition, Eq. (32) reduces to

$$\frac{[\![\mathcal{C}]\!]_\tau^2}{\langle \mathcal{C} \rangle_\tau^2} \ge \frac{1}{\mathcal{A}(\tau)}, \qquad (33)$$

which is the thermodynamic uncertainty relation derived in refs. 21,23. Therefore, Eq. (30) is a generalization of the well-known classical bounds.

## Geometric bound in quantum space

Thus far, we have considered the classical probability space. We now move to the quantum space and obtain the geometric bound for the continuous matrix product state. We consider the time evolution of $|\Psi(t)\rangle$, which is induced by the unitary in Eq. (8). We analyze the dynamics through the quantum speed limit[55]. Similar to Eq. (18), the bound for fidelity is given by the relation[45,63]:

$$\frac{1}{2} \int_{t_1}^{t_2} dt \sqrt{\mathcal{J}(t)} \ge \mathcal{L}_D(|\Psi(t_1)\rangle, |\Psi(t_2)\rangle), \qquad (34)$$

where $\mathcal{J}(t)$ is the quantum Fisher information[64]

$$\mathcal{J}(t) \equiv 4 \left[ \langle \partial_t \Psi(t) | \partial_t \Psi(t) \rangle - |\langle \partial_t \Psi(t) | \Psi(t) \rangle|^2 \right], \qquad (35)$$

and $\mathcal{L}_D$ is the Bures angle defined by

$$\mathcal{L}_D(\rho_1, \rho_2) \equiv \arccos\left[ \sqrt{\mathrm{Fid}(\rho_1, \rho_2)} \right], \qquad (36)$$

with Fid$(\rho_1, \rho_2)$ being the quantum fidelity[65]:

$$\mathrm{Fid}(\rho_1, \rho_2) \equiv \left( \mathrm{Tr}\sqrt{\sqrt{\rho_1}\rho_2\sqrt{\rho_1}} \right)^2. \qquad (37)$$

Here, $\rho_1$ and $\rho_2$ are arbitrary density operators and the fidelity satisfies $0 \le \mathrm{Fid}(\rho_1, \rho_2) \le 1$. Since $|\Psi(t)\rangle$ is a pure state, the fidelity reduces to $\mathrm{Fid}(|\Psi(t_1)\rangle, |\Psi(t_2)\rangle) = |\langle \Psi(t_2)|\Psi(t_1)\rangle|^2$. $\mathcal{L}_D$ quantifies the distance between two density operators and is widely employed in quantum speed limits[55]. Equation (34) is also commonly used in quantum speed limits[55]. The quantum Fisher information $\mathcal{J}(t)$ can be computed using the two-sided Lindblad equation introduced in ref. 66 (see Supplementary Note 3).

For the classical case, the Fisher information $\mathcal{I}(t)$ reduces to the dynamical activity $\mathcal{A}(t)$ [Eq. (23)]. However, it is difficult to represent the quantum Fisher information $\mathcal{J}(t)$ by a well-known physical quantity. Therefore, from Eq. (22), we may define the quantum generalization of the dynamical activity by

$$\mathcal{B}(t) \equiv t^2 \mathcal{J}(t), \qquad (38)$$

where the classical Fisher information $\mathcal{I}(t)$ in Eq. (22) is replaced with the quantum counterpart. In the present manuscript, we refer to $\mathcal{B}(t)$ as the quantum dynamical activity.

The fidelity obeys the monotonicity relation with respect to any completely positive and trace-preserving map[65]. Using the monotonicity, we obtain (see the Methods section for details)

$$\frac{1}{2} \int_0^\tau \frac{\sqrt{\mathcal{B}(t)}}{t} dt \ge \mathcal{L}_D(\rho(0), \rho(\tau)). \qquad (39)$$

Equation (39) is a continuous measurement case of the quantum speed limit reported in ref. 45. Regarding a quantum speed limit in open quantum dynamics, ref. 46 considered Lindblad dynamics and employed relative purity as a distance measure. Equation (39) itself can be derived from Eq. (34) via the monotonicity of the quantum fidelity. Although there are infinitely many ways to describe open quantum dynamics through purification, we will show that the quantum dynamical activity $\mathcal{B}(\tau)$ in Eq. (39) plays a central role in a quantum thermodynamic uncertainty relation derived as follows. The speed limit relations derived in Eqs. (24) and (39) do not explicitly include time $\tau$. However, by rearranging terms, we can obtain lower bounds for the evolution time $\tau$ (see the Methods section).

Next, we consider a quantum thermodynamic uncertainty relation that follows directly from Eq. (34). Again, we consider the observables $\mathcal{C}, \mathcal{C}^\circ$ and $\mathcal{C}^\bullet$. Similar to the classical case [Eq. (30)], we obtain the thermodynamic uncertainty relation for $0 \le t_1 < t_2 \le \tau$ (see the Methods section for details):

$$\left( \frac{[\![\mathcal{C}^\bullet]\!]_{t_2} + [\![\mathcal{C}^\bullet]\!]_{t_1}}{\langle \mathcal{C}^\bullet \rangle_{t_2} - \langle \mathcal{C}^\bullet \rangle_{t_1}} \right)^2 \ge \frac{1}{\tan\left[ \frac{1}{2} \int_{t_1}^{t_2} \frac{\sqrt{\mathcal{B}(t)}}{t} dt \right]^2}, \qquad (40)$$

which holds for $(1/2)\int_{t_1}^{t_2} \sqrt{\mathcal{B}(t)}/t \, dt \le \pi/2$. This relation is a quantum analog of Eq. (30) and constitutes the third result of this manuscript. Equation (40) holds for arbitrary time-independent quantum Markov processes. Similar to Eqs. (24) and (30), the quantum dynamical activity $\mathcal{B}(\tau)$ plays a fundamental role in both Eqs. (39) and (40), indicating that $\mathcal{B}(t)$ is a physically important quantity. Although we previously derived thermodynamic uncertainty relations that hold for arbitrary quantum Markov chains in refs. 17,37, the thermodynamic cost terms in refs. 17,37 are not thermodynamic quantities as in the classical case. Since Eq. (40) is the same as Eq. (30) except that $\mathcal{A}(t)$ is

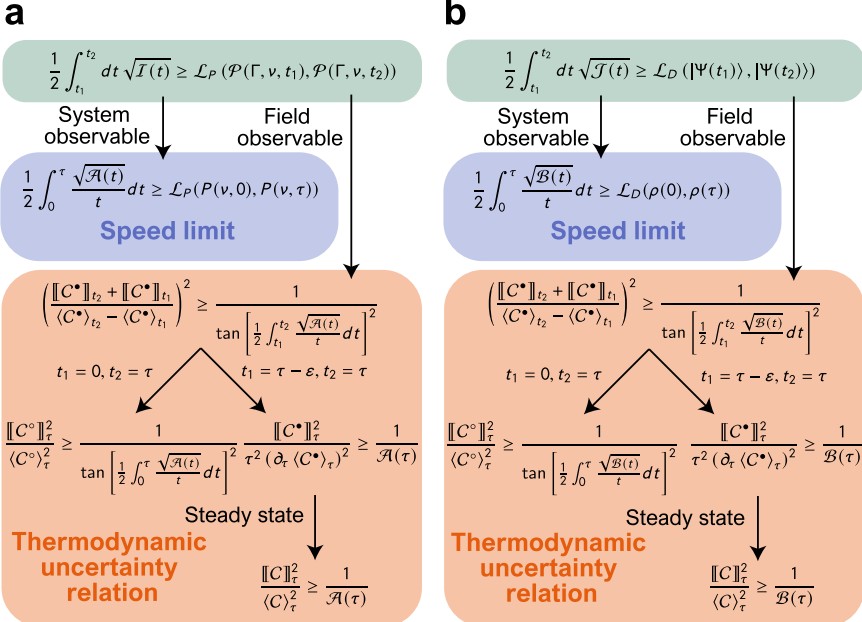

**Fig. 2 | Relation of obtained inequalities. a** When we bound the right-hand side of Eq. (18) by the system and field quantities, we obtain classical speed limits and classical thermodynamic uncertainty relations, respectively. **b** Similar relation for

the quantum case. In **a, b**, variable and function definitions are presented in Supplementary Note 8.

replaced by $\mathcal{B}(t)$, we can obtain quantum counterparts of Eqs. (31)–(33) in the same manner. Equation (31) with $\mathcal{A}(t)$ replaced by $\mathcal{B}(t)$ is a quantum thermodynamic uncertainty relation that holds for arbitrary time-independent quantum Markov processes. In particular, Eq. (33) with $\mathcal{A}(t)$ replaced by $\mathcal{B}(t)$ is equivalent to the quantum thermodynamic uncertainty relation derived in ref. 16, which was derived using the quantum Cramér–Rao inequality. In ref. 16, we calculated $\mathcal{B}(\tau)$ for $\tau \to \infty$ to show that $\mathcal{B}(\tau)$ is given by a sum of the classical dynamical activity and the coherent contribution, which is induced by the effective Hamiltonian.

In speed limits and thermodynamic uncertainty relations, the bounds require the condition $(1/2) \int_{t_1}^{t_2} \sqrt{\mathcal{A}(t)}/t \, dt \leq \pi/2$ (classical) or $(1/2) \int_{t_1}^{t_2} \sqrt{\mathcal{B}(t)}/t \, dt \leq \pi/2$ (quantum). It is helpful here to examine the physical meaning of the conditions. When the system is in a steady state, the dynamical activity is $\mathcal{A}(t) = \mathfrak{a}t$, where $\mathfrak{a}$ is a proportionality coefficient. Consequently, $(1/2) \int_0^\tau \sqrt{\mathcal{A}(t)}/t \, dt = \sqrt{\mathfrak{a}\tau}$, which transforms the constraint into $\tau \leq \pi^2/(4\mathfrak{a})$. Therefore, physically, the conditions can be identified as the constraint for $\tau$, demonstrating that the predictive power of the bounds is limited to a prescribed time determined by the system's dynamics. This limitation on $\tau$ can be ascribed to the geometric speed limit relations. In Eqs. (18) and (34), the range of values for the left-hand side is $[0, \infty)$ while that for the right-hand side is $[0, \pi/2]$. Therefore, although the geometric speed limit relations hold for $\tau \to \infty$, predictive power is lost for finite time values.

The derivations above assume the initially pure state $\rho(0) = |\psi(0)\rangle\langle\psi(0)|$. Using the purification, we can show that the speed limits and thermodynamic uncertainty relations hold for an initially mixed state (see Supplementary Note 4). Thus far, we have been concerned with the theoretical aspects of the bounds. We numerically test the speed limits and the thermodynamic uncertainty relations and verify the bounds (see Supplementary Note 5).

## Discussion
The results represented by Eqs. (24), (30), (39), and (40) show that the speed limits and the thermodynamic uncertainty relations can be understood as two different aspects of Eqs. (18) and (34). When we

bound the right-hand sides of Eqs. (18) and (34) with the quantities in the principal system, that is, the probability distribution $P(v, s)$ or the density operator $\rho(s)$, the inequalities reduce to the classical and quantum speed limits expressed by Eqs. (24) and (39), respectively. On the other hand, when we bound the right-hand sides of Eqs. (18) and (34) with the field quantity, $\langle\mathcal{C}^\bullet\rangle_t$ and $[\![\mathcal{C}^\bullet]\!]_t$, we obtain the classical and quantum thermodynamic uncertainty relations, expressed by Eqs. (30) and (40), respectively. Therefore, the speed limits and the thermodynamic uncertainty relations can be derived from the common ancestral relation. Figure 2 shows an intuitive illustration of the logical connections explained above. Note that we previously derived the classical speed limit and thermodynamic uncertainty relation in a unified way in refs. 28,67. However, refs. 28,67 derived the classical speed limit as a short time limit of the thermodynamic uncertainty relation, whereas the derivation here does not use such a distinct setting for the speed limit.

Thus far, we have considered a time-independent Markov process, meaning that $H_{\text{sys}}$ and $L_m$ are not dependent on time. Here, we examine a time-dependent case with the time-dependent operators $H_{\text{sys}}(s)$ and $L_m(s)$. It is possible to introduce a time-dependent analog of $|\Psi(t)\rangle$ introduced in Eq. (8). Using the time-dependent representation, we can derive speed limits and thermodynamic uncertainty relations similar to Eqs. (24) and (30), where the dynamical activity is replaced by the generalized dynamical activity (Supplementary Note 6).

We have considered geometric speed limit relations in the bulk space. As shown in Eq. (8), since the time evolution of the composite system comprising the system and the quantum field admits closed quantum dynamics, any relation that holds in the closed system should hold for the composite system as well. We here consider a consequence of the Heisenberg uncertainty relation[68,69], which is the most fundamental uncertainty relation in quantum mechanics, in Eq. (8). It can be shown that the Heisenberg uncertainty relation reduces to the thermodynamic uncertainty relation (Supplementary Note 7). It should also be noted that this correspondence is a consequence of the relation between the Cramér–Rao inequality and the Heisenberg uncertainty relation, as reported by ref. 70. The Heisenberg uncertainty relation is a fundamental inequality to derive the

Mandelstam-Tamm quantum speed limit[42]. Our result shows that the Heisenberg uncertainty relation also plays a fundamental role in the thermodynamic uncertainty relation when considering the bulk-boundary correspondence of the Markov process.

Thermodynamic uncertainty relations were originally derived as the inequality between current fluctuations and entropy production[19,20]. As such it might be possible to obtain a unified bulk-boundary approach for speed limits and thermodynamic uncertainty relations for which the thermodynamic cost involves solely entropy production. However, it is difficult to derive a unified bound for entropy production. To derive the bound, we should introduce another scaled continuous matrix product state that provides the same information regarding the number of jump events and the system state as the original dynamics while the Fisher information yields entropy production.

In this paper, we studied the consequences of considering the bulk-boundary correspondence in classical and quantum Markov processes. These investigations could possibly be extended to employ refined Heisenberg uncertainty relations, as shown in ref. 71, as an example. Since any uncertainty relation that holds in closed quantum dynamics should hold in the time evolution shown by Eq. (8), it can be anticipated that other uncertainty relations can be derived using the technique demonstrated herein.

## Methods

### Geometric bound

We employ the geometric bounds given by Eqs. (18) and (34) to obtain speed limits and thermodynamic uncertainty relations.

In Eq. (18), the left-hand side gives the path length corresponding to the dynamics parametrized by $t \in [0, \tau]$ that connects the two states under the Fisher information metric, while the right-hand side of Eq. (18) corresponds to the geodesic distance between the two states[61]. Similarly, in Eq. (34), the left-hand side gives the path length of the dynamics $|\Psi(t)\rangle$ under the Fubini-Study metric, while the right-hand side of Eq. (34) is the geodesic distance between the initial $|\Psi(t_1)\rangle$ and final $|\Psi(t_2)\rangle$ states under this metric.

It is helpful here to assess the uniqueness of the metrics. In probability space, except for a constant factor, the Fisher information metric is known to correspond to the unique contractive Riemannian metric. In the case that a metric in the density operator space is considered, an infinite number of metrics is possible. The geodesic distance can be analytically calculated for several metrics, such as the quantum Fisher information metric and the Wigner–Yanase information metric, both of which fall into the Fubini-Study metric for pure states. The continuous matrix product state is pure and so the Fubini-Study metric $\mathcal{J}(t)$ gives a unique metric[48].

### Number operator and observables

In the main text, we consider the observable $\mathfrak{C}(\Gamma)$ defined in Eq. (25). For the classical Markov process defined in Eq. (16), using the correspondence $m \to (\nu, \mu)$, Eq. (25) can be written as

$$\mathfrak{C}(\Gamma) = \sum_{\nu,\mu,\nu\neq\mu} \alpha_{\nu\mu} \mathfrak{N}_{\nu\mu}(\Gamma). \tag{41}$$

As an example, when $\alpha_{\mu\nu} = -\alpha_{\nu\mu}$, $\mathfrak{C}(\Gamma)$ defines the time-integrated current that is antisymmetric under time reversal. In particular, the original thermodynamic uncertainty relation[19,20] states that the fluctuation of a time-integrated current such as this is bounded from below by the reciprocal of the entropy production. In addition, if $\alpha_{\mu\nu} = -\alpha_{\nu\mu} = 1$ then $\mathfrak{C}(\Gamma)$ quantifies the amount of displacement, which can be used to quantify the elapsed time on a Brownian clock[72]. In the main text, we define $\mathcal{C}^\circ$ and $\mathcal{C}^\bullet$ in Eq. (29). When we represent these observables as functions of a trajectory $\Gamma$ as was done in

Eq. (25), we have

$$\mathfrak{C}^\circ \equiv \sum_m \alpha_m \eta_m(\mathfrak{N}_m(\Gamma)), \quad \mathfrak{C}^\bullet \equiv \sum_m \alpha_m \xi_m(\mathfrak{N}_m(\Gamma)), \tag{42}$$

where the functions $\eta_m$ and $\xi_m$ are defined in Eqs. (14) and (15), respectively. Since $\mathcal{C}^\circ$ and $\mathcal{C}^\bullet$ are generalizations of $\mathcal{C}$, they can recover $\mathcal{C}$ as a particular case. Moreover, they can express observables that are not covered by $\mathcal{C}$. An example of $\mathcal{C}^\circ$ that does not belong to $\mathcal{C}$ would be $\eta(n) = \text{sgn}(n)$, where sgn is the sign function. It gives a value of 1 when there is more than one jump but a value of 0 otherwise. We also note that $\mathcal{C}^\circ$ satisfies $\langle \mathcal{C}^\circ \rangle_{t=0} = 0$ and $[\![\mathcal{C}^\circ]\!]_{t=0} = 0$, which is an important property of $\mathcal{C}^\circ$ used in the derivation of Eq. (31).

### Derivation of speed limit relations

We derive a classical speed limit relation from Eq. (18). The Bhattacharyya coefficient satisfies monotonicity with respect to any classical channel[73]. Since $P(\nu, t) = \sum_\Gamma \mathcal{P}(\Gamma, \nu, t)$, the monotonicity yields

$$\text{Bhat}(\mathcal{P}(\Gamma, \nu, t_1), \mathcal{P}(\Gamma, \nu, t_2)) \leq \text{Bhat}(P(\nu, t_1), P(\nu, t_2)). \tag{43}$$

Substituting Eqs. (22) and (43) into Eq. (18), we obtain the classical speed limit of Eq. (24).

The quantum speed limit of Eq. (39) can be derived in a similar manner. The fidelity obeys the monotonicity relation with respect to an arbitrary, completely positive, and trace-preserving map[65]. Since $\rho(t) = \text{Tr}_{\text{fld}}\left[|\Psi(t)\rangle\langle\Psi(t)|\right]$ from Eq. (10), the following relation holds:

$$\text{Fid}(|\Psi(t_1)\rangle, |\Psi(t_2)\rangle) \leq \text{Fid}(\rho(t_1), \rho(t_2)). \tag{44}$$

Using Eqs. (44) and the quantum dynamical activity $\mathcal{B}(t)$ [Eq. (38)], we obtain Eq. (39).

### Derivation of thermodynamic uncertainty relations

Here, we derive classical thermodynamic uncertainty relations from Eq. (24). Let us consider the Hellinger distance between two probability distributions $p_1(x)$ and $p_2(x)$:

$$\begin{aligned}
\text{Hel}^2(p_1(x), p_2(x)) &\equiv \frac{1}{2}\sum_x \left(\sqrt{p_1(x)} - \sqrt{p_2(x)}\right)^2 \\
&= 1 - \text{Bhat}(p_1(x), p_2(x)).
\end{aligned} \tag{45}$$

We can assume that the probability distributions $p_1(x)$ and $p_2(x)$ are defined for a set of real values. We can define the mean and standard deviation of the distributions by $\chi_i \equiv \sum_x x p_i(x)$ and $\sigma_i \equiv \sqrt{\sum_x x^2 p_i(x) - \chi_i^2}$, respectively. Given the mean and standard deviation of $p_1(x)$ and $p_2(x)$, the lower bound of the Hellinger distance is given by[74]:

$$\text{Hel}^2(p_1(x), p_2(x)) \geq 1 - \left[\left(\frac{\chi_1 - \chi_2}{\sigma_1 + \sigma_2}\right)^2 + 1\right]^{-\frac{1}{2}}. \tag{46}$$

We previously used Eq. (46) to derive a quantum thermodynamic uncertainty relation in ref. 17. Knowing the entire trajectory $\Gamma$, we can compute the statistics of the number of jump events. As an example, for $\Gamma = [(s_1, m_1), (s_2, m_2), (s_3, m_3)]$, we know that there are three jump events at $s_1$, $s_2$, and $s_3$ during the time interval $[0, \tau]$. Therefore, according to the monotonicity of the Bhattacharyya coefficient and Eq. (46), we have

$$\text{Bhat}(\mathcal{P}(\Gamma, t_1), \mathcal{P}(\Gamma, t_2)) \leq \left[\left(\frac{\langle\mathcal{C}^\bullet\rangle_{t_1} - \langle\mathcal{C}^\bullet\rangle_{t_2}}{[\![\mathcal{C}^\bullet]\!]_{t_1} + [\![\mathcal{C}^\bullet]\!]_{t_2}}\right)^2 + 1\right]^{-\frac{1}{2}}, \tag{47}$$

where $\langle \mathcal{C}^\bullet \rangle_t$ and $[\![\mathcal{C}^\bullet]\!]_t$ are defined in Eqs. (27) and (28), respectively. For $0 \le (1/2) \int_{t_1}^{t_2} dt \sqrt{\mathcal{I}(t)} \le \pi/2$, Eq. (18) yields

$$\cos\left[\frac{1}{2}\int_{t_1}^{t_2} dt \sqrt{\mathcal{I}(t)}\right] \le \mathrm{Bhat}\big(\mathcal{P}(\Gamma,\nu,t_1), \mathcal{P}(\Gamma,\nu,t_2)\big). \quad (48)$$

Combining Eqs. (43), (47) and (48), we obtain Eq. (30).

Similarly, we can derive quantum thermodynamic uncertainty relations from Eq. (34). Regarding the quantum fidelity, a series of inequalities holds, as were employed in ref. 17:

$$
\begin{aligned}
|\langle \Psi(t_2)|\Psi(t_1)\rangle| &\le \sum_\Gamma |\langle \Psi(t_2)|\Gamma\rangle\langle\Gamma|\Psi(t_1)\rangle| \\
&\le \sum_\Gamma \sqrt{\mathcal{P}(\Gamma,t_1)\mathcal{P}(\Gamma,t_2)} \quad (49) \\
&= \mathrm{Bhat}\big(\mathcal{P}(\Gamma,t_1),\mathcal{P}(\Gamma,t_2)\big).
\end{aligned}
$$

The triangle inequality is used in the first line, while the Cauchy–Schwarz inequality is employed in the first to second lines. From Eq. (34), for $0 \le (1/2)\int_{t_1}^{t_2} dt \sqrt{\mathcal{J}(t)} \le \pi/2$, we have

$$\cos\left[\frac{1}{2}\int_{t_1}^{t_2} dt \sqrt{\mathcal{J}(t)}\right] \le |\langle\Psi(t_2)|\Psi(t_1)\rangle|. \quad (50)$$

Combining Eqs. (47), (49) and (50), we can derive Eq. (40).

Next, we derive the conventional thermodynamic uncertainty relation, which was derived in ref. 23, from Eq. (30). We consider a time interval $[\tau - \varepsilon, \tau]$ for Eq. (30), where $\varepsilon > 0$ is an infinitesimally small parameter. Then we obtain

$$\left(\frac{[\![\mathcal{C}^\bullet]\!]_\tau + [\![\mathcal{C}^\bullet]\!]_{\tau-\varepsilon}}{\langle\mathcal{C}^\bullet\rangle_\tau - \langle\mathcal{C}^\bullet\rangle_{\tau-\varepsilon}}\right)^2 \ge \frac{1}{\tan\left[\frac{1}{2}\int_{\tau-\varepsilon}^\tau \frac{\sqrt{\mathcal{A}(t)}}{t}dt\right]^2}. \quad (51)$$

Since $\varepsilon$ is sufficiently small, we have

$$\frac{d\langle\mathcal{C}^\bullet\rangle_t}{dt} = \frac{\langle\mathcal{C}^\bullet\rangle_t - \langle\mathcal{C}^\bullet\rangle_{t-\varepsilon}}{\varepsilon}. \quad (52)$$

Moreover, we consider a perturbation expansion for $[\![\mathcal{C}^\bullet]\!]_{\tau-\varepsilon}$:

$$[\![\mathcal{C}^\bullet]\!]_{\tau-\varepsilon} = [\![\mathcal{C}^\bullet]\!]_\tau + \varepsilon b_1 + \varepsilon^2 b_2 + \cdots, \quad (53)$$

where $b_i \in \mathbb{R}$ are expansion coefficients. Since $\varepsilon \ll 1$, considering the Taylor expansion $(\tan x)^2 = x^2 + O(x^3)$, we obtain

$$
\begin{aligned}
\tan\left[\frac{1}{2}\int_{\tau-\varepsilon}^\tau \frac{\sqrt{\mathcal{A}(t)}}{t}dt\right]^2 &\simeq \left(\frac{1}{2}\int_{\tau-\varepsilon}^\tau \frac{\sqrt{\mathcal{A}(t)}}{t}dt\right)^2 \quad (54) \\
&= \frac{\mathcal{A}(\tau)}{4\tau^2}\varepsilon^2.
\end{aligned}
$$

Substituting Eqs. (52)–(54) into Eq. (51), we obtain

$$\left(\frac{2[\![\mathcal{C}^\bullet]\!]_\tau + \varepsilon b_1 + \varepsilon^2 b_2 + \cdots}{\varepsilon \partial_\tau \langle\mathcal{C}^\bullet\rangle_\tau}\right)^2 \ge \frac{4\tau^2}{\mathcal{A}(\tau)\varepsilon^2}. \quad (55)$$

Taking a limit of $\varepsilon \to 0$, we obtain Eq. (32). We can repeat the same calculation for the quantum dynamical activity $\mathcal{B}(t)$.

### Speed limit relation as minimum evolution time
Speed limit relations are often provided as bounds for the minimum evolution time. As detailed in ref. 75, from speed limit relations shown in Eqs. (24) and (39), we can introduce two types of minimum evolution time. These can be explained using the quantum bound [Eq. (39)] because ref. 75 addressed a quantum speed limit relation. The first type

of minimum evolution time $\tau_{\min}$ can be implicitly defined by

$$\frac{1}{2}\int_0^{\tau_{\min}} \frac{\sqrt{\mathcal{B}(t)}}{t} dt = \mathcal{L}_D(\rho(0), \rho(\tau)). \quad (56)$$

Here, $\tau_{\min}$ is the time required to reach the geodesic length between $\rho(0)$ and $\rho(\tau)$ traveling along the actual evolution path.

The second type of minimum evolution time can be obtained directly from Eq. (39). Let us define the average evolution speed as follows:

$$\mathcal{V}_{av} \equiv \frac{1}{\tau}\int_0^\tau dt \frac{\sqrt{\mathcal{B}(t)}}{2t}. \quad (57)$$

Using $\mathcal{V}_{av}$, we obtain the bound:

$$\tau \ge \tau_{av} \equiv \frac{\mathcal{L}_D(\rho(0), \rho(\tau))}{\mathcal{V}_{av}}. \quad (58)$$

Note that the evaluation of Eq. (58) requires information regarding $\tau$ because $\mathcal{V}_{av}$ is typically dependent on $\tau$. When considering a unitary evolution induced by a time-independent Hamiltonian and pure states, $\tau_{\min} = \tau_{av}$ holds, but they do not agree in general dynamics. Note that $\tau_{\min}$ and $\tau_{av}$ can be defined in the classical bound [Eq. (24)] in the same manner.

## Data availability
The data generated in this study are provided in the Source Data file. Source data are provided with this paper.

## Code availability
All codes used in this study are available from https://github.com/yoshihiko-hasegawa/BulkBoundaryBounds.

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

## Acknowledgements
The fruitful comments of Tan Van Vu are greatly appreciated. This work was supported by JSPS KAKENHI Grant Numbers JP19K12153 and JP22H03659.

## Author contributions
This work was carried out by Y.H.

## Competing interests
The author declares no competing interests.
