## [Peer Review File · Nature Communications]

REVIEWER COMMENTS

Reviewer #1 (Remarks to the Author):

In this paper the Author shows a deep and interesting connection between quantum speed limits and the so-called Thermodynamic Uncertainty Relations.

Both these two topics have attracted a significant attention over the past years, due to their interdisciplinary and fundamental nature as well as their and far-reaching applications. Showing an explicit link between them is therefore a major achievement, which deserves publication in Nat. Comm.

Furthermore, the method used to bridge these two concepts takes its roots from the bulk-boundary correspondence, a powerful technique widely employed e.g. in conformal field theory but extremely uncommon within stochastic thermodynamics.

In light of the novelty of the results and of the methodologies, I therefore recommend this paper for publication after revision and clarifications of the following points:

1. While I totally understand the introduction of the scaled quantum field, I wonder whether Eq.9 is the unique solution that would produce the desired result. Can the author comment on this? On this note, would it be possible to employ a different metric than the Bures one and therefore not have the need to compute the fidelity with respect to same-time density matrices (thus avoiding the introduction of the scaled field)?
2. It seems to me that Eq. 14, and thus the following conclusions, should also be valid for observables that commute with the total number operator. Is this the case? If so, the Author should emphasize this fact which would strengthen his results even further.
3. The Author should make more explicit the connection between Eq 26 and the geometric speed limit in Ref 47. In particular it may not be immediately clear how to extract the time τ from here, invert the relation and then translate Eq 26 into a lower bound on τ (which would be a speed limit bound).
4. It is not clear to me what are the physical and dynamical implications of the constraint below Eq. 30 and 36, i.e. $\frac{1}{2} \int_0^\tau dt \frac{\sqrt{A(t)}}{t} \leq \pi/2$. This upper bound on the integrated dynamical activity, which is crucial to the validity of both these central inequalities (30 and 36) should be more precisely and thoroughly discussed: does it constraint the underlying dynamics? Is it physical? How

stringent is it? This also relates with the following point, which is what I feel is the only critique I have about this work.

5. The paper presents a beautiful and deep theoretical result, but it also lacks a clearcut example or application. This could be helpful to grasp more clearly the implications (also in relation to point 3) and the paper would greatly benefit from such an addition.

6. I did not find particularly compelling the section titled "Bound for two-point measurement". The reason is that the Author proceeds to show that Eq 57 holds true only under the assumption that the system initially starts in an eigenstate of the measured observable, which therefore effectively turns the two-point measurement scheme effectively to a single point one. Furthermore, I do not think this distinction is even particularly needed, since it is known that the statistics of a two-point measurement scheme is equivalent to the one in terms of jump trajectories in the case of weak-coupling and Lindblad dynamics (see e.g. <https://doi.org/10.48550/arXiv.cond-mat/0703347>) which is a crucial starting assumption of this whole work too. I would maybe suggest moving this Section to the Appendix.

Reviewer #2 (Remarks to the Author):

The manuscript reports the derivation of fundamental relations for common quantum kinetics based on the Bulk-Boundary correspondence. This is a topic of high interest for the community and definitely suits the criteria of Nature communications. However, I feel that the present manuscript is very difficult to access for a typical researcher in the field. Without going through heavy math it is impossible to identify the main findings and judge the applicability of the research: Thus I suggest to strongly modify the manuscript along the following points:

- 1) Clearly specify at an early stage (with equations) the relations for Speed limit and thermodynamic uncertainty relation, which shall be proven. Not all readers are aware of them.
- 2) Focus on the main points rather than addressing related aspects of the approach (such as relativistic issues not considered here)
- 3) Provide a summary stating which equations are the main outcome. Something like: The main result is the proof of Eq.(1111), which is
- 4) Move derivations to methods/appendix while only a skeleton of relevant assumptions, definitions and results remain in the main text. I had actually huge difficulties to understand Fig 2 as the definitions of numerous symbols was difficult to track.

Reviewer #3 (Remarks to the Author):

Thermodynamic speed limits and uncertainty relations have received plenty of attention in recent years, mainly due to the development of non-equilibrium stochastic thermodynamics. In this contribution, the author shows how different speed limits and uncertainty relations can be obtained from geometric speed limits. This is done by encoding the history of the quantum or classical stochastic evolution of a given system into a quantum field. When the geometric speed limit is applied to the reduced state encoding the instantaneous state of the system, the author recovers known versions of speed limits. When the geometric speed limit is applied to the quantum field encoding the history of the evolution, the authors recovers known expressions for thermodynamic uncertainty relations, in addition to apparently unknown ones (Eqs. (52) and (53)).

I like the article and I think it is a relevant contribution to an active research field. However, I think it is extremely technical in nature, and therefore more appropriate to a specialized journal. I appreciate the fact that the author draws many interesting analogies between different quantities pertaining to different fields, and in my opinion there is where the value of the work mainly lies. On the other hand, the main practical result of this work seems to be a novel technique to obtain known results (with the possible exception of Eqs. (52) and (53)). Also, the connection between this manuscript and the bulk-boundary correspondence in AdS/CFT seems a little far fetched to me.

On a more technical note, I am surprised that nowhere in the article it is mentioned the notion of entropy production. The reason is that most versions of speed limits and thermodynamic uncertainty relations I am aware of are expressed in terms of the entropy production (for example, the original observation of TURs by Seifert in PRL 114, 158101 (2015), or recent results like Phys. Rev. Lett. 127, 160601). This is related to the fact that in its original formulation, thermodynamic uncertainty relations apply to observables that are odd under time-reversal. However, in the present manuscript, uncertainty relations seem to not be related to time-reversal, and the relevant quantity is the dynamical activity instead of the entropy production. It would be nice if the author could comment on this difference. In particular, what would it take to recover the original version of the thermodynamic uncertainty relation using the presented formalism?

As a final comment, while I think that the writing of the article is sufficiently clear, I also think there is plenty of room for improvement. I often had the feeling the writing was repetitive. I also think that the article would benefit if the main results are stated at the beginning in the more non-technical way possible, leaving their derivation for later.

As I said before I feel that this article, as it is written, belongs to a more specialized journal.

Dear Dr. Omelchenko and Referees

I have considered the referees' comments and criticisms and I am hereby resubmitting the revised manuscript to *Nature Communications*. Below, I have presented the comments and questions of the referees in italic typeface, followed by my responses. The equation and figure numbers included only in this response are prefixed with the letter "R" [for example, Eq. (R1) or Fig. R1]. Numbers without a prefix [for example, Eq. (1) or Fig. 1] refer to those in the main text. Please note that, due to changes in the structure of the revised manuscript, the equation and reference numbers presented in the revised version are different from those in the original version. Because I have copied the referees' comments verbatim, the equation and reference numbers in these comments correspond to those in the original paper.

The following changes have been made to the main text:

1. Simplification of the derivation in the main text and addition of subsections in the Methods providing the detailed derivation.
2. Removal of topics that are not directly related to the main findings.
3. Generalisation of counting observables used in the thermodynamic uncertainty relations.
4. Addition of the physical meaning of the constraint included in the thermodynamic uncertainty relations.
5. Text addition to better identify the main findings of the work.
6. Removal of many repetitive sentences.
7. Comment on relations between entropy production and the thermodynamic bounds.
8. Addition of a subsection showing how the speed limit relations can be related to the minimum evolution time.

In addition, the following changes have been made to the supplementary information:

9. Addition of numerical simulations for the speed limit relations and thermodynamic uncertainty relations [Supplementary Note 5].
10. Addition of function and variable definitions for Fig. 2 [Supplementary Note 8].

1 First referee

In this paper the Author shows a deep and interesting connection between quantum speed limits and the so-called Thermodynamic Uncertainty Relations.

Both these two topics have attracted a significant attention over the past years, due to their interdisciplinary and fundamental nature as well as their and far-reaching applications. Showing an explicit link between them is therefore a major achievement, which deserves publication in Nat. Comm. Furthermore, the method used to bridge these two concepts takes its roots from the bulk-boundary correspondence, a powerful technique widely employed e.g. in conformal field theory but extremely uncommon within stochastic thermodynamics. In light of the novelty of the results and of the methodologies, I therefore recommend this paper for publication after revision and clarifications of the following points:

Reply

I would like to thank the referee for assessing the manuscript and for providing constructive comments. These suggestions are much appreciated and have improved the quality of the manuscript. Below are my responses to the referee's comments.

1.1

While I totally understand the introduction of the scaled quantum field, I wonder whether Eq.9 is the unique solution that would produce the desired result. Can the author comment on this? On this note, would it be possible to employ a different metric than the Bures one and therefore not have the need to compute the fidelity with respect to same-time density matrices (thus avoiding the introduction of the scaled field)?

Reply

For a classical probability space, the Fisher information metric represents the unique contractive Riemannian metric. In contrast, in the case of a density operator space, there are an infinite number of contractive Riemannian metrics. The most well-known is the Bures metric, whose geodesic length is given by the Bures angle. The Wigner-Yanase information metric is an alternative to the Bures metric. However, as stated in Ref. (Pires et al., 2016), the Fubini-Study metric is the unique contractive metric in quantum pure state space. Since the continuous matrix product state of Eq. (8) is a pure state, the Fubini-Study metric is the unique metric for this state. In accordance with this comment, I have added the following text to the revised manuscript.

Page 8, left column, line 4

It is helpful here to assess the uniqueness of the metrics. In probability space, below a specific factor, the Fisher information metric is known to correspond to the unique contractive Riemannian metric. Even so, in the case that a metric in the density operator space is considered, an infinite number of metrics is possible. The geodesic distance can be analytically calculated for several metrics, such as the quantum Fisher information metric and the Wigner-Yanase information metric, both of which fall into the Fubini-Study metric for pure states. The continuous matrix product state is pure and so the Fubini-Study metric $\mathcal{J}(t)$ gives a unique metric [48].

I would also like to comment on the uniqueness of the scaled quantum field. In this work, the scaled field was required to satisfy two conditions. Specifically, the scaled matrix product state had to provide the correct density operator, $\rho(\tau)$, and both the unscaled and scaled matrix product states had to yield the same statistics on the total number operator. Although I was not able to rigorously prove that the scaled representation [Eq. (8)] employed in the manuscript is the only possible means of realizing the abovementioned conditions, it appears unlikely that these conditions could be achieved with another scaling.

1.2

It seems to me that Eq. 14, and thus the following conclusions, should also be valid for observables that commute with the total number operator. Is this the case? If so, the Author should emphasize this fact which would strengthen his results even further.

Reply

I thank the referee for indicating the possibility of more general observables.

As the referee indicated, generalized observables that commute with the number operator can be employed in thermodynamic uncertainty relations. Let \mathcal{N}_m be the total number operator having the eigendecomposition:

$$\mathcal{N}_m = \sum_{n_m=0} n_m |n_m\rangle \langle n_m|. \quad (\text{R1})$$

Here, n_m is the number of m th jump events detected within $[0, \tau]$ and $|n_m\rangle$ is the corresponding state. Consider the following generalization of \mathcal{N}_m :

$$\mathcal{N}_m^\circ \equiv \sum_{n_m=0} \eta_m(n_m) |n_m\rangle \langle n_m|, \quad (\text{R2})$$

where $\eta_m(n)$ is a real function satisfying $\eta_m(0) = 0$. This zero condition is required when deriving Eq. (31). Moreover, we can define the generalization as follows:

$$\mathcal{N}_m^\bullet \equiv \sum_{n_m=0} \xi_m(n_m) |n_m\rangle \langle n_m|. \quad (\text{R3})$$

Here, $\xi_m(n)$ is any real function. Using \mathcal{N}_m° and \mathcal{N}_m^\bullet , we can also define the observables:

$$\mathcal{C}^\circ \equiv \sum_m \alpha_m \mathcal{N}_m^\circ, \quad \mathcal{C}^\bullet \equiv \sum_m \alpha_m \mathcal{N}_m^\bullet. \quad (\text{R4})$$

The difference between Eqs. (R2) and (R3) is that $\eta_m(n)$ demands the condition $\eta_m(0) = 0$ while $\xi_m(n)$ does not. I found that the observables in Eqs. (30) and (32) can be generalized to \mathcal{C}^\bullet while the observable in Eq. (31) can be generalized to \mathcal{C}° . Note that the bounds satisfied by \mathcal{C}° and \mathcal{C}^\bullet should hold for the original observable \mathcal{C} as well.

In the revised version of the paper, I have added the following text.

Page 4, left column, line 4

Thus far, our focus has been on the number operator \mathcal{N}_m alone, but more general observables can be considered. The number operator admits the eigendecomposition:

$$\mathcal{N}_m = \sum_{n_m=0} n_m |n_m\rangle \langle n_m|, \quad (13)$$

where the eigenvalue n_m denotes the number of m th jumps within $[0, \tau]$ and $|n_m\rangle$ is its corresponding state. The first-level generalization of \mathcal{N}_m is

$$\mathcal{N}_m^\circ \equiv \sum_{n_m=0} \eta_m(n_m) |n_m\rangle \langle n_m|, \quad (14)$$

where $\eta_m(n)$ is a real function satisfying $\eta_m(0) = 0$. Thus, \mathcal{N}_m° is a generalization of \mathcal{N}_m as $\eta_m(n) = n$ recovers \mathcal{N}_m in Eq. (13). The second-level generalization would be

$$\mathcal{N}_m^\bullet \equiv \sum_{n_m=0} \xi_m(n_m) |n_m\rangle \langle n_m|, \quad (15)$$

where $\xi_m(n)$ is an arbitrary real function. \mathcal{N}_m^\bullet is the most general form of observable that commutes with \mathcal{N}_m . Note that \mathcal{N}_m° and \mathcal{N}_m^\bullet can also be used for the scaled representation $|\Psi(t)\rangle$ instead of $|\Phi(t)\rangle$ (see the Methods section).

1.3

The Author should make more explicit the connection between Eq 26 and the geometric speed limit in Ref 47. In particular it may not be immediately clear how to extract the time tau from here, invert the relation and then translate Eq 26 into a lower bound on tau (which would be a speed limit bound).

Reply

In accordance with the referee's comment, I have made the connection between speed limits in my paper and that in the indicated reference more explicit.

There are two types of minimum evolution time associated with the geometric quantum speed limit (Mirkin et al., 2016). The geometric quantum speed limit is given by

$$\mathcal{L}_D(\rho(0), \rho(\tau)) \leq \int_0^\tau dt \frac{\sqrt{\mathcal{J}(t)}}{2}. \quad (R5)$$

The first type of minimum evolution time τ_{\min} is implicitly defined by

$$\mathcal{L}_D(\rho(0), \rho(\tau)) = \int_0^{\tau_{\min}} dt \frac{\sqrt{\mathcal{J}(t)}}{2}. \quad (R6)$$

Here, τ_{\min} is the time required to reach the geodesic length traveling along the actual evolution path. With regard to the second type of minimum evolution time, we can define the average speed of the evolution as

$$\mathcal{V}^{\text{av}} \equiv \frac{1}{\tau} \int_0^\tau \frac{\sqrt{\mathcal{J}(t)}}{2} dt, \quad (R7)$$

which is the time average of the right hand side of Eq. (R5). Then, from Eq. (R5), we proceed as follows:

$$\mathcal{L}_D(\rho(0), \rho(\tau)) \leq \tau \times \frac{1}{\tau} \int_0^\tau dt \frac{\sqrt{\mathcal{J}(t)}}{2} \quad (R8)$$

$$= \tau \mathcal{V}^{\text{av}}, \quad (R9)$$

yielding the following bound:

$$\tau \geq \frac{\mathcal{L}_D(\rho(0), \rho(\tau))}{\mathcal{V}^{\text{av}}} \equiv \tau^{\text{av}}. \quad (\text{R10})$$

Note that \mathcal{V}^{av} does not correspond to the actual evolution speed. Several speed limit relations employ the form of Eq. (R10), which is a consequence of the obvious identity [Eq. (R8)]. However, it is widely recognized that the bound of Eq. (R10) is problematic in that \mathcal{V}^{av} itself depends on τ .

Since the speed limit relations derived in my paper are based on the geometric quantum speed limit, both of the above definitions can be used. In accordance with the referee's comment, I have added the following text to the revised version.

Page 6, right column, line 11

The speed limit relations derived in Eqs. (24) and (39) do not explicitly include time τ . However, by rearranging terms, we can obtain lower bounds for the evolution time τ (see the Methods section).

Additionally, the following subsection has been appended to the Methods section.

Page 10, left column, line 11

Speed limit relations are often provided as bounds for the minimum evolution time. As detailed in Ref. [76], from speed limit relations shown in Eqs. (24) and (39), we can introduce two types of minimum evolution time. These can be explained using the quantum bound [Eq. (39)] because Ref. [76] addressed a quantum speed limit relation. The first type of minimum evolution time τ_{min} can be implicitly defined by

$$\frac{1}{2} \int_0^{\tau_{\text{min}}} \frac{\sqrt{\mathcal{B}(t)}}{t} dt = \mathcal{L}_D(\rho(0), \rho(\tau)). \quad (56)$$

Here, τ_{min} is the time required to reach the geodesic length between $\rho(0)$ and $\rho(\tau)$ traveling along the actual evolution path.

The second type of minimum evolution time can be obtained directly from Eq. (39). Let us define the average evolution speed as follows:

$$\mathcal{V}_{\text{av}} \equiv \frac{1}{\tau} \int_0^{\tau} dt \frac{\sqrt{\mathcal{B}(t)}}{2t}. \quad (57)$$

Using \mathcal{V}_{av} , we obtain the bound:

$$\tau \geq \tau_{\text{av}} \equiv \frac{\mathcal{L}_D(\rho(0), \rho(\tau))}{\mathcal{V}_{\text{av}}}. \quad (58)$$

Note that the evaluation of Eq. (58) requires information regarding τ because \mathcal{V}_{av} is typically dependent on τ . When considering a unitary evolution induced by a time-independent Hamiltonian and pure states, $\tau_{\text{min}} = \tau_{\text{av}}$ holds but they do not agree in general dynamics. Note that τ_{min} and τ_{av} can be defined in the classical bound [Eq. (24)] in the same manner.

1.4

It is not clear to me what are the physical and dynamical implications of the constraint below Eq. 30 and 36, i.e. $1/2 \int_0^{\tau} dt \frac{\sqrt{\mathcal{A}(t)}}{t} \leq \pi/2$. This upper bound on the integrated dynamical activity, which is crucial to the validity of both these central inequalities (30 and 36) should be more precisely and thoroughly discussed: does it constraint the underlying dynamics? Is it physical? How stringent is it? This also relates with the following point, which is what I feel is the only critique I have about this work.

Reply

The indicated condition is given by

$$\frac{1}{2} \int_0^\tau dt \frac{\sqrt{\mathcal{A}(t)}}{t} \leq \frac{\pi}{2}. \quad (\text{R11})$$

Here, I would like to elaborate on this condition. Suppose that the system is near the steady state, meaning that $\mathcal{A}(t)$ can be approximated as

$$\mathcal{A}(t) \simeq \mathbf{a}t, \quad (\text{R12})$$

where \mathbf{a} is a proportionality coefficient. Substituting Eq. (R12) into Eq. (R11) and integrating reduces Eq. (R11) to

$$\tau \lesssim \frac{\pi^2}{4\mathbf{a}}. \quad (\text{R13})$$

Therefore, physically, the condition of Eq. (R11) can be identified as the constraint for τ . This shows that the predictive power of the bound is limited to a specific time determined by the proportionality coefficient \mathbf{a} . The condition of Eq. (R13) can be ascribed to the ancestral classical or quantum geometric speed limit relations. The geometric speed limit [Eq. (18) in the main text] is

$$\frac{1}{2} \int_0^\tau dt \sqrt{\mathcal{I}(t)} \geq \arccos [\text{Bhat}(\mathcal{P}(\Gamma, \nu, 0), \mathcal{P}(\Gamma, \nu, \tau))]. \quad (\text{R14})$$

The range of values for the left hand side of this equation is $[0, \infty)$, while that of the right hand side is $[0, \pi/2]$. In addition, the left hand side goes to infinity for $\tau \rightarrow \infty$. This indicates that the predictive power of this equation is lost at a finite time although Eq. (R14) is valid for $\tau \rightarrow \infty$. Therefore, the geometric speed limit relation implicitly has a constraint regarding range of τ values.

The following text has been added to the revised manuscript.

Page 6, right column, line 7 from the bottom

In speed limit and thermodynamic uncertainty relations, the bounds require the condition $(1/2) \int_{t_1}^{t_2} \sqrt{\mathcal{A}(t)}/t dt \leq \pi/2$ (classical) or $(1/2) \int_{t_1}^{t_2} \sqrt{\mathcal{B}(t)}/t dt \leq \pi/2$ (quantum). It is helpful here to examine a physical meaning of the conditions. When the system is in a steady state, the dynamical activity is $\mathcal{A}(t) = \mathbf{a}t$, where \mathbf{a} is a proportionality coefficient. Consequently, $(1/2) \int_0^\tau \sqrt{\mathcal{A}(t)}/t dt = \sqrt{\mathbf{a}\tau}$, which transforms the constraint into $\tau \leq \pi^2/(4\mathbf{a})$. Therefore, physically, the conditions can be identified as the constraint for τ , demonstrating that the predictive power of the bounds is limited to a prescribed time determined by the system's dynamics. This limitation on τ can be ascribed to the geometric speed limit relations. In Eqs. (18) and (34), the range of values for the left hand side is $[0, \infty)$ while that for the right hand side is $[0, \pi/2]$. Therefore, although the geometric speed limit relations hold for $\tau \rightarrow \infty$, predictive power is lost for finite time values.

1.5

The paper presents a beautiful and deep theoretical result, but it also lacks a clearcut example or application. This could be helpful to grasp more clearly the implications (also in relation to point 3) and the paper would greatly benefit from such an addition.

Reply

Based on this comment, I performed numerical experiments regarding both the speed limit and thermodynamic uncertainty relations using a N_S state Markov process (classical) and a two level atom driven by a classical laser field (quantum). Due to the length limitation imposed by *Nature Communications*, these results have been appended in Supplementary Note 5. Figures S2 and S3 present the results of these calculations. These numerical simulations demonstrate the following:

- Verification of classical speed limit of Eq. (24) [Figs. S2a and b]. Figures S2a and b are based on different initial distributions, which shows how the dynamics saturates the bound.

- Verification of quantum speed limit of Eq. (39) [Figs. S2c and d]. Figures S2c and d highlight the effects of quantumness on the bound.
- Verification of classical and quantum thermodynamic uncertainty relations, shown by Eq. (31) and its quantum counterpart, in an out-of-steady state condition [Figs. S3a and c].
- Violation of conventional steady-state thermodynamic uncertainty relation, shown by Eq. (33) and its quantum counterpart, in an out-of-steady state condition [Figs. S3b and d]. In Figs. S3b and d, some points are under the solid line, indicating that Eq. (33) and its quantum counterpart do not hold under the out-of-steady state condition.

1.6

I did not find particularly compelling the section titled “Bound for two-point measurement” . The reason is that the Author proceeds to show that Eq 57 holds true only under the assumption that the system initially starts in an eigenstate of the measured observable, which therefore effectively turns the two-point measurement scheme effectively to a single point one. Furthermore, I do not think this distinction is even particularly needed, since it is known that the statistics of a two-point measurement scheme is equivalent to the one in terms of jump trajectories in the case of weak-coupling and Lindblad dynamics (see e.g. <https://doi.org/10.48550/arXiv.cond-mat/0703347>)which is a crucial starting assumption of this whole work too. I would maybe suggest moving this Section to the Appendix.

Reply

As the referee indicated, the eigenstate assumption for the initial state provides a significant restriction since the first measurement has no effect. The paper recommended by the referee indicates that the statistics of fluctuations associated with the two-point measurements and the trajectory-based approach agree under the assumptions. Because I would prefer for the paper to focus on the main points of the research, I have removed this section from the revised manuscript.

2 Second referee

The manuscript reports the derivation of fundamental relations for common quantum kinetics based on the Bulk-Boundary correspondence. This is a topic of high interest for the community and definitely suits the criteria a of Nature communications. However, I feel that the present manuscript is very difficult to access for a typical researcher in the field. Without going through heavy math it is impossible to identify the main findings and judge the applicability of the research: Thus I suggest to strongly modify the manuscript along the following points:

Reply

I would like to thank the referee for reviewing the manuscript and providing helpful comments, which I feel have improved the readability of the paper. Below are my responses to the individual comments.

2.1

Clearly specify at an early stage (with equations) the relations for Speed limit and thermodynamic uncertainty relation, which shall be proven. Not all readers are aware of them.

Reply

Based on this comment, I have added text to the Introduction that explains the main purpose of the manuscript, as follows.

Page 1, right column, line 2 from the bottom

We apply the concept of the geometric speed limit inequality to the bulk space to derive speed limits [Eqs. (24) and (39)] and thermodynamic uncertainty relations [Eqs. (30) and (40)]. In the resulting speed limit relations, the distances between the initial and the final states are

bounded from above by terms comprising classical or quantum dynamical activities. In the case of the thermodynamic uncertainty relations obtained in this work, we show that the precision of an observable that counts the number of jumps is bounded from below by costs composed of classical or quantum dynamical activities. We establish a duality relation in that the speed limit and the thermodynamic uncertainty relation can be understood as two different aspects of the geometric speed limit inequality.

2.2

Focus on the main points rather than addressing related aspects of the approach (such as relativistic issues not considered here)

Reply

In response to this comment, I have removed topics that are not directly related to the main findings. Specifically, I have removed the relativistic descriptions and the paragraph that explains AdS/CFT.

2.3

Provide a summary stating which equations are the main outcome. Something like: The main result is the proof of Eq.(1111), which is

Reply

Additional text has been added to better identify the main findings of the work. Specifically, the following text has been incorporated in the revised version of the paper.

Page 5, left column, line 5

Equation (24) is the first result of this paper, showing that the distance between the initial and final probability distributions in a classical Markov process has an upper bound comprising the dynamical activity $\mathcal{A}(t)$.

Page 5, right column, line 9

Equation (30) is the second result of this paper and holds for an arbitrary time-independent classical Markov process.

Page 6, right column, line 23

This relation is a quantum analog of Eq. (30) and constitutes the third result of this manuscript.

2.4

Move derivations to methods/appendix while only a skeleton of relevant assumptions, definitions and results remain in the main text. I had actually huge difficulties to understand Fig 2 as the definitions of numerous symbols was difficult to track.

Reply

Based on this suggestion, the structure of the manuscript has been thoroughly revised such that all of the derivations are now included in the Methods section. Specifically, the following subsections have been created in the Methods section to describe details of the derivations.

- Geometric bound
- Number operator and observables

- Derivation of speed limit relations

I believe that readers will now be able to understand the main findings of the work without going into the technical details.

Regarding the problem of lack of definitions in Fig. 2, I have added a new section titled “Supplementary Note 8: Function and variable definitions” to the Supplementary Information, which is also provided below. In addition, text has been added to the caption of Fig. 2 to inform readers of the definitions of the variables and functions that are summarized in Supplementary Note 8.

Supplementary Note 8

$\mathcal{I}(t)$ Fisher information [Eq. (19)]

$\mathcal{J}(t)$ Quantum Fisher information [Eq. (35)]

Γ Trajectory [Eq. (3)]

$\mathcal{P}(\Gamma, \nu, t)$ Probability of observing trajectory Γ and being Y_ν at time t [Eq. (17)]

$|\Psi(t)\rangle$ Scaled continuous matrix product state at time t [Eq. (8)]

$P(\nu, t)$ Probability of ν at time t [Eq. (16)]

$\rho(t)$ Density matrix at time t [Eq. (1)]

$\mathcal{A}(t)$ Dynamical activity [Eq. (23)]

$\mathcal{B}(t)$ Quantum dynamical activity [Eq. (38)]

\mathcal{L}_P Bhattacharya angle [Eq. (20)]

\mathcal{L}_D Bures angle [Eq. (36)]

\mathcal{C} Weighted sum of the number operator \mathcal{N}_m [Eqs. (13) and (26)]

\mathcal{C}° Weighted sum of the number operator \mathcal{N}_m° [Eqs. (14) and (29)]

\mathcal{C}^\bullet Weighted sum of the number operator \mathcal{N}_m^\bullet [Eqs. (15) and (29)]

3 Third referee

Thermodynamic speed limits and uncertainty relations have received plenty of attention in recent years, mainly due to the development of non-equilibrium stochastic thermodynamics. In this contribution, the author shows how different speed limits and uncertainty relations can be obtained from geometric speed limits. This is done by encoding the history of the quantum or classical stochastic evolution of a given system into a quantum field. When the geometric speed limit is applied to the reduced state encoding the instantaneous state of the system, the author recovers known versions of speed limits. When the geometric speed limit is applied to the quantum field encoding the history of the evolution, the authors recovers known expressions for thermodynamic uncertainty relations, in addition to apparently unknown ones (Eqs. (52) and (53)).

I like the article and I think it is a relevant contribution to an active research field. However, I think it is extremely technical in nature, and therefore more appropriate to a specialized journal. I appreciate the fact that the author draws many interesting analogies between different quantities pertaining to different fields, and in my opinion there is where the value of the work mainly lies. On the other hand, the main practical result of this work seems to be a novel technique to obtain known results (with the possible exception of Eqs. (52) and (53)). Also, the connection between this manuscript and the bulk-boundary correspondence in AdS/CFT seems a little far fetched to me.

Reply

I would like to thank the referee for critical reading of the manuscript and for providing several suggestions that have improved the quality of the paper. Below are my point-by-point responses to the referee’s

comments.

Regarding the issue of the technical complexity of the paper, I have moved the detailed technical derivations to the Methods section or to Supplementary Information. Based on this change, the core concept of this manuscript should be understandable to the reader without going into technical details. Specifically, I have added the following subsections to the Methods section.

- Geometric bound
- Number operator and observables
- Derivation of speed limit relations

Only the definitions, results and a brief proof are now provided in the main part of the manuscript.

I would also like to address the comment that this work illustrates a novel technique for obtaining known results (with some exceptions). This paper places emphasis on the finding that speed limit relations and thermodynamic uncertainty relations can be identified as two aspects of the same ancestral geometric bound. Moreover, in addition to this conceptual finding, the inequalities derived in this work are themselves of importance. The previous thermodynamic uncertainty relation (Di Terlizzi & Baiesi, 2019) that holds true for an out-of-steady state system is given by:

$$\frac{\llbracket \mathcal{C} \rrbracket_\tau^2}{\tau^2 (\partial_\tau \langle \mathcal{C} \rangle_\tau)^2} \geq \frac{1}{\mathcal{A}(\tau)}. \quad (\text{R15})$$

This equation is problematic in that its left hand side cannot be interpreted as the precision, which is the central quantity in thermodynamic uncertainty relations. Furthermore, because the left hand side of this relationship involves the derivative ∂_τ it is difficult to ascertain its validity with numerical or physical experiments, as the derivative is vulnerable to noise. In contrast, the thermodynamic uncertainty relation derived in my paper is

$$\frac{\llbracket \mathcal{C} \rrbracket_\tau^2}{\langle \mathcal{C} \rangle_\tau^2} \geq \frac{1}{\tan \left[\frac{1}{2} \int_0^\tau \frac{\sqrt{\mathcal{A}(t)}}{t} dt \right]^2}. \quad (\text{R16})$$

Equation (R16) also holds for out-of-steady state cases and does not suffer from the challenges associated with Eq. (R15) described above.

With regard to the concern that the relationship between the manuscript and AdS/CFT is far-fetched, I would first like to clarify the terminologies used in this paper, as follows.

Bulk/boundary correspondence A phenomenon in which all the information of the bulk space is encoded in its boundary.

AdS/CFT correspondence A scenario in which the CFT at the boundary is equivalent to the AdS in the bulk space. Several quantities (such as viscosity and entanglement entropy) in CFT can be calculated by solving the gravity equation for the bulk AdS space.

These definitions indicate that the bulk/boundary correspondence incorporates a wider range of concepts than the AdS/CFT correspondence. That is, AdS/CFT is one particular aspect of bulk-boundary correspondence. This work was based on the general concept of bulk/boundary correspondence, which is why the title includes this phrase. In particular, the boundary state encodes the temporal jump processes and the extra dimension functions as a scaling of the boundary. It should be noted that a prior publication (Osborne et al., 2010) that proposed a continuous matrix product state included the phrase “holographic quantum states” in its title. Here, the term “holographic” is associated with the holographic principle, which in turn is based on bulk/boundary correspondence. Moreover, although the bulk/boundary correspondence employed in this manuscript is not equivalent to AdS/CFT correspondence, there are some similarities between the two concepts. For these reasons, I believe it is reasonable to state that the methodology outlined in the present manuscript is related to bulk/boundary correspondence in stochastic processes.

Having said that, I agree that the original paper contained some potentially misleading expressions that could lead the reader to assume that the technique employed in this manuscript was based on the AdS/CFT correspondence. In the revised paper, a paragraph that explains the AdS/CFT correspondence has been removed.

3.1

On a more technical note, I am surprised that nowhere in the article it is mentioned the notion of entropy production. The reason is that most versions of speed limits and thermodynamic uncertainty relations I am aware of are expressed in terms of the entropy production (for example, the original observation of TURs by Seifert in PRL 114, 158101 (2015), or recent results like Phys. Rev. Lett. 127, 160601). This is related to the fact that in its original formulation, thermodynamic uncertainty relations apply to observables that are odd under time-reversal. However, in the present manuscript, uncertainty relations seem to not be related to time-reversal, and the relevant quantity is the dynamical activity instead of the entropy production. It would be nice if the author could comment on this difference. In particular, what would it take to recover the original version of the thermodynamic uncertainty relation using the presented formalism?

Reply

This comment relates to the thermodynamic cost included in the speed limits and thermodynamic uncertainty relations. As noted in the above reply, the main point of this paper was to unify the speed limits and thermodynamic uncertainty relations. Consequently, the manuscript shows that these two uncertainty relations have the same ancestral inequality. The primary consequence of this ancestral inequality is that the cost terms of the speed limit and the thermodynamic uncertainty relations are identical. In particular, the classical speed limit and the thermodynamic uncertainty relations are represented by

$$\frac{1}{2} \int_0^\tau \frac{\sqrt{\mathcal{A}(t)}}{t} dt \geq \mathcal{L}_P(P(\nu, 0), P(\nu, \tau)), \quad (\text{R17})$$

$$\frac{\langle\langle \mathcal{C} \rangle\rangle_\tau^2}{\langle \mathcal{C} \rangle_\tau^2} \geq \frac{1}{\tan \left[\frac{1}{2} \int_0^\tau \frac{\sqrt{\mathcal{A}(t)}}{t} dt \right]^2}. \quad (\text{R18})$$

Note that both bounds include $\frac{1}{2} \int_0^\tau \frac{\sqrt{\mathcal{A}(t)}}{t} dt$ (shown in red text). From this, it should be possible to employ a similar discussion with regard to entropy production to unify the speed limit and thermodynamic uncertainty relations. It is well known that the thermodynamic cost in the conventional thermodynamic uncertainty relation (Barato & Seifert, 2015) is given by entropy production. However, it is unlikely that a speed limit relation can be bounded by entropy production *alone*. To examine this, we can designate $P(\nu, s)$ as the probability distribution at time s for a classical Markov process. As detailed in Ref. (Dechant, 2022), which bound the probability distribution using the Wasserstein distance, it is possible to realize a desired time evolution $P(\nu, s)$ with vanishingly small entropy production. This finding indicates that, in the case that we consider the distance between the initial and final probability distributions in a Markov process, it is not possible to generate the bound based on entropy production alone. As noted above, the primary purpose of this paper was to demonstrate a unification of the two uncertainty relations. However, if we were to use entropy production alone as the thermodynamic cost, it would not be possible to describe speed limit and thermodynamic uncertainty relations in a unified manner.

I would also like to comment on the differences between the thermodynamic uncertainty relations for dynamical activity and entropy production. As indicated by the referee, the entropy-production thermodynamic uncertainty relation considers time-integrated currents that are antisymmetric under time-reversal. Conversely, observables handled by the dynamical-activity thermodynamic uncertainty relation do not require this condition and so are more general than the entropy-production counterparts.

In accordance with this comment, the following has been added to the Discussion section.

Page 7, right column, line 22

Thermodynamic uncertainty relations were originally derived as the inequality between current fluctuations and entropy production [19, 20]. As such it might be possible to obtain a unified bulk/boundary approach for speed limit and thermodynamic uncertainty relations for which the thermodynamic cost involves solely entropy production. However, this is unlikely because the desired time evolution $P(\nu, s)$ can be realized with vanishingly small entropy production [71]. Therefore, considering a speed limit with respect to entropy production alone yields a meaningless bound.

In addition, with regard to the observable associated with the entropy-production thermodynamic uncertainty relation, I have added the following text to the Methods section.

Page 8, right column, line 15 from the bottom

As an example, when $\alpha_{\mu\nu} = -\alpha_{\nu\mu}$, $\mathfrak{C}(\Gamma)$ defines the time-integrated current that is antisymmetric under time reversal. In particular, the original thermodynamic uncertainty relation [19, 20] states that the fluctuation of a time-integrated current such as this is bounded from below by the reciprocal of the entropy production.

3.2

As a final comment, while I think that the writing of the article is sufficiently clear, I also think there is plenty of room for improvement. I often had the feeling the writing was repetitive. I also think that the article would benefit if the main results are stated at the beginning in the more non-technical way possible, leaving their derivation for later.

Reply

On the basis of this comment, I have revised the paper to remove many of the repetitive sentences. Moreover, I have removed the equations related to quantum thermodynamic uncertainty relations that were identical to the classical counterparts except for the dynamical activity term. The quantum thermodynamic uncertainty relation part is now described as follows.

Page 6, right column, line 20 from the bottom

Since Eq. (40) is the same as Eq. (30) except that $\mathcal{A}(t)$ is replaced by $\mathcal{B}(t)$, we can obtain quantum counterparts of Eqs. (31)–(33) in the same manner. Equation (31) with $\mathcal{A}(t)$ replaced by $\mathcal{B}(t)$ is a quantum thermodynamic uncertainty relation that holds for arbitrary time-independent quantum Markov processes. In particular, Eq. (33) with $\mathcal{A}(t)$ replaced by $\mathcal{B}(t)$ is equivalent to the quantum thermodynamic uncertainty relation derived in Ref. [16]. In Ref. [16], we calculated $\mathcal{B}(\tau)$ for $\tau \rightarrow \infty$ to show that $\mathcal{B}(\tau)$ is given by a sum of the classical dynamical activity and the coherent contribution, which is induced by the effective Hamiltonian.

Some theoretical papers initially present the main equation and then provide its derivation later. However, because my paper contains four main equations, it was difficult to place these four equations in the early part of the manuscript. Instead, I have added text to the Introduction that explains the main findings and their meanings in a nontechnical manner, as follows.

Page 1, right column, line 2 from the bottom

We apply the concept of the geometric speed limit inequality to the bulk space to derive speed limits [Eqs. (24) and (39)] and thermodynamic uncertainty relations [Eqs. (30) and (40)]. In the resulting speed limit relations, the distances between the initial and the final states are bounded from above by terms comprising classical or quantum dynamical activities. In the case of the thermodynamic uncertainty relations obtained in this work, we show that the precision of an observable that counts the number of jumps is bounded from below by costs composed of classical or quantum dynamical activities. We establish a duality relation in that the speed limit and the thermodynamic uncertainty relation can be understood as two different aspects of the geometric speed limit inequality.

References

- Barato, A. C., & Seifert, U. (2015). Thermodynamic uncertainty relation for biomolecular processes. *Phys. Rev. Lett.*, *114*, 158101. <https://doi.org/10.1103/PhysRevLett.114.158101>
- Dechant, A. (2022). Minimum entropy production, detailed balance and Wasserstein distance for continuous-time Markov processes. *J. Phys. A: Math. Theor.*, *55*, 094001. <https://doi.org/10.1088/1751-8121/ac4ac0>

- Di Terlizzi, I., & Baiesi, M. (2019). Kinetic uncertainty relation. *J. Phys. A: Math. Theor.*, *52*, 02LT03. <https://doi.org/10.1088/1751-8121/aace34>
- Mirkin, N., Toscano, F., & Wisniacki, D. A. (2016). Quantum-speed-limit bounds in an open quantum evolution. *Phys. Rev. A*, *94*, 052125. <https://link.aps.org/doi/10.1103/PhysRevA.94.052125>
- Osborne, T. J., Eisert, J., & Verstraete, F. (2010). Holographic quantum states. *Phys. Rev. Lett.*, *105*, 260401. <https://doi.org/10.1103/PhysRevLett.105.260401>
- Pires, D. P., Cianciaruso, M., Céleri, L. C., Adesso, G., & Soares-Pinto, D. O. (2016). Generalized geometric quantum speed limits. *Phys. Rev. X*, *6*, 021031. <https://doi.org/10.1103/PhysRevX.6.021031>

REVIEWERS' COMMENTS

Reviewer #1 (Remarks to the Author):

In the revised manuscript, the Author answered meaningfully and satisfactorily to all the questions and comments made by myself and by the other Referees, implementing meaningful changes and decisively improving the (already high) quality of the paper.

The added simulations and plots in Supplementary Note 5, although involving rather simple and well-known examples, neatly provide clearcut evidence of the validity of the newly derived bounds.

The plot S3, in particular, is compelling; what I would only recommend is that the Author should expand a bit (also by providing some references) on the violations of TUR for the out-of-steady-state situation (which he briefly alludes to in reference of subplots 3b) and 3d). The reader might in fact not be familiar with all the literature on TUR and would benefit from a more articulate explanation / set of references.

There is finally probably a typo in the line 7 of the Caption of Fig S3, where it should be "In (d)" rather than "In (b)" (i.e. the reference to the subplots should be corrected).

I therefore recommend this paper for publication in Nature Communications with those few minor changes.

Reviewer #2 (Remarks to the Author):

The author has answered the issues raised by the reviewers very well. This lead to a significant improvement of the manuscript. I have one more concern regarding the statement

"Because systems handled by stochastic and quantum thermodynamics are far from equilibrium, such systems can be described by a Markov process that is correlated and coupled."

on page 1. I do not see, why being far from equilibrium supports the validity of the Markov approximation which is commonly a practical necessity with quite severe restrictions. Maybe the author wanted to say something different.

Reviewer #3 (Remarks to the Author):

The author has successfully addressed my early comments and I am happy to recommend the acceptance of this article in Nature Communications.

Dear Dr. Omelchenko and Referees

I have considered the referees' comments and I am hereby resubmitting the revised manuscript to *Nature Communications*. Below, I have presented the comments and questions of the referees in italic typeface, followed by my responses.

The following changes have been made to the main text:

1. I have revised the sentence in the introduction to address the comment of the second referee and to ensure that it accurately reflects the intended meaning [page 1, left column, line 18].
2. In the previous version, the projector in Eqs. (13)-(15) implicitly assumed that they are rank 1, i.e., $|n_m\rangle\langle n_m|$. Since this assumption does not necessarily hold, they were changed to arbitrary rank projectors Π_{n_m} [Eqs. (13)-(15) in page 3]. This revision will not affect any other parts of the paper.
3. In the previous version, I wrote "the desired time evolution $P(\nu, s)$ can be realized with vanishingly small entropy production" in the discussion. However, this statement turned out to be inaccurate because this is true for systems that allow time-dependent transition rates (i.e., time-dependent jump operators). This sentence has been corrected in the revised version [page 7, left column, line 12 from the bottom].

In addition, the following change has been made to the supplementary information:

1. I have added text to provide further explanation of TURs in out-of-steady-state situations, addressing comment from the first referee [page 7, line 6 and line 3 from the bottom].

I also attached the manuscript with revisions highlighted in magenta, which include the above-mentioned modifications and other minor corrections.

1 First referee

In the revised manuscript, the Author answered meaningfully and satisfactorily to all the questions and comments made by myself and by the other Referees, implementing meaningful changes and decisively improving the (already high) quality of the paper.

The added simulations and plots in Supplementary Note 5, although involving rather simple and well-known examples, neatly provide clearcut evidence of the validity of the newly derived bounds. The plot S3, in particular, is compelling; what I would only recommend is that the Author should expand a bit (also by providing some references) on the violations of TUR for the out-of-steady-state situation (which he briefly alludes to in reference of subplots 3b) and 3d). The reader might in fact not be familiar with all the literature on TUR and would benefit from a more articulate explanation / set of references. There is finally probably a typo in the line 7 of the Caption of Fig S3, where it should be "In (d)" rather than "In (b)" (i.e. the reference to the subplots should be corrected.

I therefore recommend this paper for publication in Nature Communications with those few minor changes.

Reply

I would like to express my gratitude for your recommendation of acceptance after the revisions and for providing me with constructive feedback that is invaluable in improving the quality of my work. Following the comments of the referee, I have added text in the revised Supplementary Material that provides further explanation of TURs in out-of-steady-state situations. Moreover, the indicated typo in the Caption of Fig. S3 has been fixed in the revised version.

2 Second referee

The author has answered the issues raised by the reviewers very well. This lead to a significant improvement of the manuscript. I have one more concern regarding the statement "Because systems handled by stochastic and quantum thermodynamics are far from equilibrium, such systems can be described by a Markov process that is correlated and coupled." on page 1. I do not see, why being far from equilibrium supports the validity of the Markov approximation which is commonly a practical necessity with quite severe restrictions. Maybe the author wanted to say something different.

Reply

I would like to express my gratitude for providing valuable feedback that has been instrumental in improving the quality of my work. As the referee indicated, the sentence in the introduction did not accurately convey the intended meaning. I have revised the sentence to ensure that it now clearly communicates the intended message. The sentence now becomes “Stochastic and quantum thermodynamic systems exhibit behaviors that occur far from equilibrium and are described by correlated and coupled Markov processes. This fact leads us to consider that the bulk-boundary correspondence might play a fundamental role in stochastic and quantum thermodynamics.” The sentence in the introduction has been modified accordingly.

3 Third referee

The author has successfully addressed my early comments and I am happy to recommend the acceptance of this article in Nature Communications.

Reply

Your recommendation of accepting my paper is greatly appreciated. I appreciate your time and effort in carefully reviewing my paper that is invaluable in improving the quality of my work.